# Dating Hiatuses: A Statistical Model of the Recent Slowdown in Global Warming – and the Next One

J. Isaac Miller[1] and Kyungsik Nam[1]

[1]University of Missouri

**Correspondence:** J. Isaac Miller (millerjisaac@missouri.edu)

**Abstract.** Much has been written about the so-called hiatus or pause in global warming, also known as the stasis period, the start of which is typically dated to 1998. HadCRUT4 global mean temperatures slightly decreased over 1998-2013, though a simple statistical model predicts that they should have grown by $0.016°$C/yr, in proportion to the increases in concentrations of well-mixed greenhouse gases and ozone. We employ a statistical approach to assess the contributions of model forcings and natural variability to the hiatus. Our point estimates suggest that none of the model forcings explain more than a third of the missing heat, accounting for the upper bound of the confidence interval on the effect of tropospheric aerosols, which is the most prominent yet most uncertainly measured of the model forcings that could explain the missing heat. The El Niño Southern Oscillation (ENSO) explains up to about a third of the missing heat, and two thirds and possibly up to 81% is explained by the unusually high temperature of 1998. Looking forward, the simple model also fails to explain the large increases since then ($0.087°$C/year over 2013-2016). This period coincides with another El Niño, but the ENSO fails to satisfactorily account for the increase. We propose instead a semiparametric cointegrating statistical model that augments an energy balance model with a novel multibasin measure of the oceans' multidecadal temperatures cycles. The model partially explains the recent slowdown and explains all of the subsequent warming. The natural cycle suggests the possibility – depending in part on the rate of increase of WMGHG concentrations – of a much longer hiatus over roughly 2023-2061, with potentially important implications for policy evaluation.

## 1 Introduction

There is a well-established physical and statistical link between temperatures and anthropogenic and natural climate forcings. A simple linear cointegrating regression of the HadCRUT4 global mean temperature anomaly (GMT) onto the radiative forcings given by Hansen *et al.* (2017) explains $88\%$ of the variation in mean temperature using variations in these forcings. Constraining all but volcanic forcings to have a common coefficient in the regression explains $84\%$.

Over the period of 1998-2013, the second regression, estimated using a canonical cointegrating regression, predicts an increase of $0.239°$C or $0.016°$C/yr on average, in proportion to the increase in well-mixed greenhouse gases (WMGHGs) and ozone over this period. Instead, observed GMT slightly *decreased* by $0.024°$C or $0.002°$C/yr on average, earning this period the nicknames of the "stasis period," or "hiatus" or "pause" in global warming. The difference, measured in this way as $0.263°$C or $0.018°$C/yr, is the so-called "missing heat" of the hiatus, which is quite substantial in the context of the aggregate

temperature increase since the pre-industrial era of $0.85°C$ (IPCC, 2013). In contrast, over 2013-2016, temperatures increased by $0.087°C$/yr, much *faster* than this simple statistical model predicts.

Our notion of hiatus is roughly consistent with that of Meehl *et al.* (2011), Kosaka and Xie (2013), and Drijfhout *et al.* (2014), who reference the apparent hiatus in global warming with respect to heat flux from greenhouse gases or model forcings more generally. Instead, some authors refer to the hiatus with respect to temperature changes or a trend over an earlier period (Schmidt *et al.*, 2014; Karl *et al.*, 2015; Yao *et al.*, 2016; and Medhaug *et al.*, 2017), while some authors refer to the hiatus without any explicit baseline. Linking missing heat to contemporaneous model forcings is physically appealing, and our empirical evidence suggests that our measure of missing heat comes from a cointegrating regression,[1] so the approach is statistically appealing, too. A slowdown or hiatus in global warming as we have defined it does not require a similar slowdown in forcings. On the contrary, such a hiatus is defined in spite of continuing increases in WMGHG concentrations.

What caused this hiatus? Various studies attribute it to one or more of (a) natural variability of the ocean cycles, particularly the Atlantic Multidecadal Oscillation (AMO), the Pacific Decadal Oscillation, and the El Niño Southern Oscillation (ENSO) (Kosaka and Xie, 2013; Steinman *et al.*, 2015; Yao *et al.*, 2016); (b) cooling from stratospheric aerosols released by volcanic activity (Vernier *et al.*, 2011; Neely *et al.*, 2013); (c) variability of the solar cycle (Huber and Knutti, 2014); (d) a change in the oceans' heat uptake and a weakening of the thermohaline circulation, particularly the Atlantic Meridional Overturning Circulation (AMOC) (Meehl *et al.*, 2011; Drijfhout *et al.*, 2014; Chen and Tung, 2014, 2016); (e) increased anthropogenic emissions of sulfur dioxide from bringing online a large number of coal-burning power plants in China (Kaufmann *et al.*, 2011); and (f) coverage bias or poor data more generally (Cowtan and Way, 2014; Karl *et al.*, 2015). Schmidt *et al.* (2014), Pretis *et al.* (2015), and Medhaug *et al.* (2017) emphasize the need to account for multiple explanations for the hiatus.

Keenlyside *et al.* (2008) note an acceptance in the literature that the AMO and multidecadal periodic global temperature fluctuations more generally are related to the AMOC. Drijfhout *et al.* (2014) posit a physical link through changes in heat uptake across multiple oceans. Specifically, a weakening/strengthening of AMOC, meaning less/more convection and turbulent heat loss, leads to increased/decreased net heat uptake and therefore lower/higher surface temperatures. The change in net heat uptake occurs across multiple ocean basins, so we label the resulting multidecadal temperature cycle the Oceanic Multidecadal Oscillation (OMO). In other words, the OMO is the global sea surface temperature fluctuation resulting from changes in the thermohaline circulation. The OMO contrasts with the AMO in scope – the latter is defined for the North Atlantic – but the two may be highly correlated with similar periodicities.

We propose a new method to measure the OMO, recognizing the possibility of heterogeneous long-run effects of anthropogenic forcings on ocean basins and allowing for a multibasin contribution to global mean temperatures, in the spirit of Drijfhout *et al.* (2014) and Wyatt and Curry (2014). The method allows an improvement over the linear detrending method of Enfield *et al.* (2001) or a single ocean approach such as the AMO signal estimated by Trenberth and Shea (2006). Not only do we estimate the mean OMO, but we also estimate a global distribution representing the contribution of the OMO to spa-

---

[1]The regression of temperature anomalies on volcanic forcings, the sum of the remaining forcings, and an intercept yields a covariance stationary residual series. Augmented Dickey-Fuller tests with lag lengths up to four reject the null of no cointegration. Estimates of the memory parameter of about 0.49 using a lag truncation parameter up to 10 suggest the possibility that the residual series is stationary with long memory, which also supports (fractional) cointegration.

tially disaggregated sea surface anomalies. In doing so, we carefully decompose the distribution of temperature anomalies into components with long memory, low-frequency, stochastic trending behavior (mapped to forcings from WMGHGs), with short memory, medium-frequency, cyclical behavior (the OMO), and with short memory, high-frequency, idiosyncratic behavior.

We utilize a semiparametric cointegration statistical approach (Park *et al.*, 2010), with widely used and publicly available data sets to estimate an energy balance model (EBM) similar to the well-known model of North (1975) and North and Cahalan (1981). The estimated OMO enters the model nonparametrically, as does the quasi-periodic southern oscillation index (SOI), a common proxy for the ENSO. However, information criteria select a linear specification for the latter.

We find that the solar cycle and multidecadal ocean cycle have warmed rather than cooled over the period 1998-2013, so they cannot account for the missing heat. Volcanoes, tropospheric aerosols & surface albedo, and ENSO account for about $1\%$, $19\%$, and $24\%$ respectively of the missing heat of the hiatus. The upper bounds on the uncertainty intervals for the latter two are both about one third. An even simpler explanation – that the hiatus is defined by starting in an unusually warm year, even taking into account El Niño – explains about two thirds of the missing heat, a result that echoes previous authors (Medhaug *et al.*, 2017, e.g.). A model that takes into account all but the residual just mentioned explains about $42\%$.

Roberts *et al.* (2015) speculate that the hiatus could last through the end of the decade, and Chen and Tung (2014) and Knutson *et al.* (2016) make stronger statements about its continuation. If so, then the unusually warm years of 2015-2016 are outliers and global temperatures can be expected to stabilize or cool in the next decade. On the contrary, our proposed model explains nearly all of the more recent record warm years, overshooting the record high anomaly of $0.773°C$ in 2016 by less than $0.001°C$. This result provides conclusive statistical evidence that the hiatus is over. In other words, we date the end of the recent hiatus prior to 2015.

Can we expect a future hiatus or slowdown? If so, when? We find that the two most influential non-seasonal drivers of global aggregate temperature change are the long-run contribution of WMGHGs and the fairly predictable OMO with a period of 76 years, consistent with the 65-80 year period estimated for the AMO in the literature (Knight *et al.*, 2005; Trenberth and Shea, 2006; Keenlyside *et al.*, 2008; Gulev *et al.*, 2013; Wyatt and Curry, 2014). Although the OMO cannot explain the recent hiatus, it can explain past multidecadal cooling or hiatus periods, such as the decades following the temperature spikes in about 1877 and 1943.

Kaufmann *et al.* (2006a) note about the period 1944-76 that the decrease in net radiative forcing resulting from an increase in anthropogenic sulfur emissions approximately offset the increase in net forcing from an increase in WMGHGs. Aside from the uncertainty in measuring forcing from sulfur emissions, this offset nicely shows the relationship between the OMO and temperatures: while the OMO declined by an average of $0.016°C$ per year, temperature also declined by an average of $0.012°C$ per year. The decline in temperature over this period may be explained both by a decline in the OMO and an increase in sulfur emissions.

If we condition the model on future forcings with growth rates similar to RCP8.5, we can expect temperatures to increase with a possible slowdown but without any future hiatus. However, if we condition on future forcings with the same average annual rate as that of the past 76 years, similarly to RCP6.0, we expect a multidecadal hiatus over approximately 2023-2061.

Note that our finding of a warm period separating the previous slowdown from the next one is exactly consistent with the recent projection of warming from 2018-2022 by Sévellec and Drijfhout (2018) using a different model and method.

## 2  Empirical Results

We utilize forcings data over 1850-2016 from Hansen *et al.* (2017).[2] We create two radiative forcing series: the sum of forcings from well-mixed greenhouse gases (primarily $CO_2$, $CH_4$, $N_2O$, and CFCs), ozone, tropospheric aerosols & surface albedo, and solar irradiance, denoted by $h_1$, and that from volcanoes, denoted by $h_2$. Shindell (2014) suggests the possibility that forcings due to aerosols and ozone may have effects that are different from those of WMGHGs. By aggregating all non-volcanic forcings into $h_1$, we are instead following Estrada *et al.* (2013), Pretis (2015), *inter alia*. A Wald test shows no statistically significant difference ($p$ value of $0.61$) between models with and without the restriction imposed.[3]

Some authors, such as Estrada *et al.* (2013), ignore volcanoes in statistical estimation of EBMs. Leaving out volcanoes is statistically justified by the apparent uncorrelatedness of this series with the other forcings. Relegating that series to the error term may affect statistical uncertainty, but should not bias the estimates of the effect of $h_1$. Because one of our goals is to assess the impact of volcanic activity on the slowdown, we include volcanoes. However, we allow for a separate coefficient on $h_2$, in order to accommodate the suggestion of Lindzen and Giannitsis (1998) of a smaller sensitivity parameter for physical models that include volcanic forcing.

We use HadCRUT4 and HadSST3 temperature anomaly data, measured relative to 1961-1990, from Morice *et al.* (2012) and Kennedy *et al.* (2011a,b) respectively.[4] In order to estimate the global distribution of the OMO, monthly HadSST3 data observed over $5°$ latitude by $5°$ longitude are pooled into years over 1850-2016 (167 years of data). The HadCRUT4 data set combines HadSST3 for sea and CRUTEM4 for land, so the temperatures from HadCRUT4 and HadSST3 are comparable. However, using only HadSST3 for the distribution ensures that grid boxes containing both land and ocean stations will contain only ocean measurements in the distribution.

The SOM (Supplementary Online Material) contains a detailed description of the methodology used to estimate the distribution of the OMO, the probability density function of which we denote by $f_t(r)$ for year $t$ with support $[r^-, r^+]$ (see the bottom panel of Figure S.3). Our methodology omits both long-run temporal temperature trends to avoid cointegration with $h_t$ and idiosyncratic noise to avoid over-fitting very short-run fluctuations in GMT using sea-surface temperatures. It is intuitively similar to band-pass filters used to identify cycles in economic and other oscillating time series, but it is not executed in the frequency domain.

---

[2]Annual data for 1850-2015 downloaded from www.columbia.edu/~mhs119/Burden on May 15, 2017. See the Supplementary Online Matrial (SOM) for details on the extrapolation to 2016.

[3]The test is executed as $qF$, where $q = 3$ is the number of restrictions tested and $F$ is the F-test of these restrictions based on Cochrane-Orcutt transformed regressions to accommodate an AR(1) error consistent with the bootstrapping strategy discussed below. The $q = 3$ restrictions are that the effects on temperature of a W/m$^2$ change in well-mixed greenhouse gases, ozone, tropospheric aerosols & surface albedo, and solar irradiance are equal.

[4]Ensemble median of HadCRUT.4.5.0.0 (annual unsmoothed globally averaged) and HadSST.3.1.1.0 (monthly globally disaggregated) downloaded from www.metoffice.gov.uk/hadobs on July 18, 2017 and April 4, 2017 respectively.

Consistent with an energy balance model, filtering long-run stochastic trends (low frequency on the spectrum) is accomplished by regressing out the anthropogenic signal (see Figure S.1 of the SOM). In contrast, linear detrending (Enfield *et al.*, 2001) unnecessarily assumes a constant growth rate of the anthropogenic signal. More modern approaches, such as that of Trenberth and Shea (2006) and Lenton *et al.* (2017) also filter the anthropogenic signal, but they do so indirectly as a difference

between temperature series both subjected to the same stochastic trend.

Detrending results in a noisy oscillation. High-frequency filtering could be accomplished using spectral methods if the time series were long enough or if the noise were "quiet" enough. However, it is difficult to identify a cycle with a 65-80 year period from a time series of 167 years in the presence of high-frequency cycles with substantial amplitudes, such as ENSO and the solar cycle. By focusing on a single periodic function, we narrow the desired frequency band enough to identify the cycle and

to estimate it very transparently in the time domain. Specifically, we fit the stochastically detrended temperatures to a single periodic function (see Table S.1 and Figure S.4 of the SOM).

We base our statistical model on an EBM given by

$$T^a = h^{*\prime}\alpha + \int_{r^-}^{r^+} b(r)f(r)dr + c(S) + \eta, \tag{1}$$

where $T^a$ is the global mean temperature anomaly (GMT), $h^* = (1, h_1, h_2)'$ is global forcing, $S$ is the Southern Oscillation

Index (SOI, Ropelewski and Jones, 1987) used to proxy ENSO quasi-periodic cycles,[5] $\alpha = (\alpha_0, \alpha_1, \alpha_2)'$ is a coefficient vector, and $\eta$ is an error term.

A detailed derivation of the EBM from a more familiar EBM similar to those of North (1975), North and Cahalan (1981), North *et al.* (1981), *inter alia* is provided in the SOM. A primary intuition for the derivation is that the nonlinear functions $b(r)$ and $c(S)$ allow the oceans' heat uptake to vary over multidecadal and interannual oscillations.

In order to estimate the EBM in (1) nonparametrically in $b(r)$ and $c(S)$, we attach time subscripts and write

$$T_t^a = h_t^{*\prime}\alpha + x_t'\gamma + w_t'\delta + \eta_t, \tag{2}$$

where $x_t = (x_{1t}, ..., x_{m_T t})' = \int_{r^-}^{r^+} b_{1:m_T}(r)f_t(r)dr$ with $b_{1:m_T}(r) = (b_1(r), ..., b_{m_T}(r))'$ and $w_t = (w_{1t}, ..., w_{m_S t})' = c_{1:m_S}(S_t)$ with $c_{1:m_S}(S_t) = (c_1(S_t), ..., c_{m_S}(S_t))'$, finite-order series approximation to $\int_{r^-}^{r^+} b(r)f_t(r)dr$ and $c(S_t)$, with $m_S$- and $m_T$-vectors of coefficients given by $\gamma$ and $\delta$ respectively. The error term $\eta_t$ contains both (serially correlated) stochastic forcings, along the lines of North *et al.* (1981), and any approximation errors from the series approximations.

### 2.1   The 1998-2013 Episode

The missing heat of the recent hiatus is defined above by the difference between the actual GMT in 2013 and the temperature predicted from increases in WMGHG and Ozone (G+Z, hereafter) alone using the restricted model with $\gamma, \delta = 0$ and starting in 1998. The GMT in 1998 was $0.536°C$. Fixing 1998 as the starting year and based on an increase of climate forcings from

G+Z of $0.561$ W/m$^2$ over 1998-2013, the model predicts a GMT of $0.536 + 0.561 \times 0.426 \simeq 0.775°C$ in 2013, where $0.426$

---

[5]Southern Oscillation Index downloaded from www.esrl.noaa.gov/psd/gcos_wgsp/ on April 13, 2018.

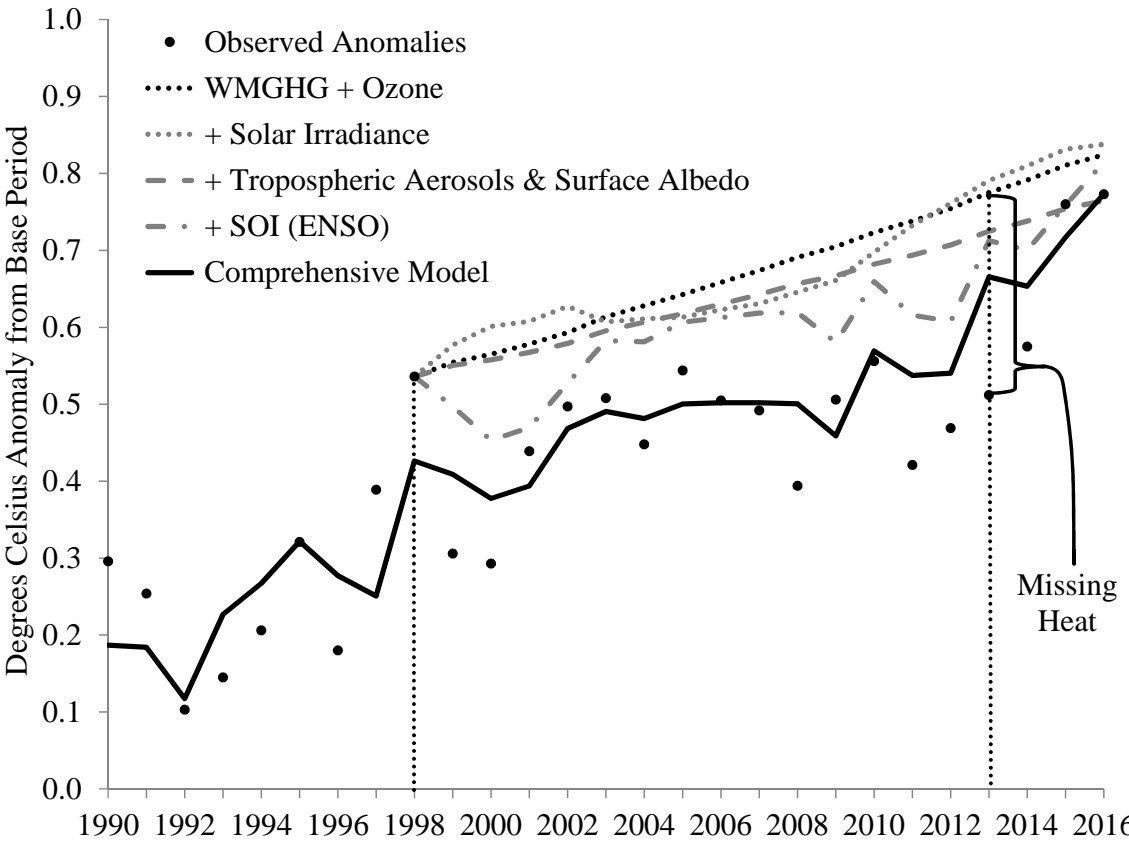

**Figure 1. A Visual Anatomy of the 1998-2013 Episode.** The hiatus is defined by the missing heat in 2013 relative to that predicted by increases in WMGHG and ozone forcings since 1998. Plots crossing the missing heat help to explain it (pictured: tropospheric aerosols & surface albedo, ENSO, and the comprehensive model), while those passing above the missing heat exacerbate it (pictured: solar irradiance).

is the CCR estimate of $\alpha_1$ in the model in (2) with $\gamma, \delta = 0$ (see Table S.2 of the SOM), with a 90% confidence interval of $(0.756, 0.795)^{\circ}$C.[6] In contrast, the observed GMT is $0.512^{\circ}$C in 2013, so that the difference, $0.775 - 0.512 \simeq 0.263^{\circ}$C $(0.244, 0.283)^{\circ}$C, represents the missing heat. The 1998-2013 episode is illustrated by the missing heat in Figure 1. Figure 2 shows the contributions of key potential explanations of the hiatus as a percentage of the missing heat with 90% confidence intervals.

One way to try to explain the missing heat is to "turn on" some of the other forcings in the model. To that end, we estimate the model in (2) with both $\gamma, \delta \neq 0$ (unrestricted) and $\gamma, \delta = 0$ (restricted). Least squares is expected to be consistent, but we use the canonical cointegrating regression approach of Park *et al.* (2010) in order to estimate the coefficients asymptotically

---

[6]The intervals throughout the paper are given with 90% confidence, in keeping with those for the forcings given by the IPCC (Myhre *et al.*, 2013). The intervals reflect not only statistical uncertainty from the regression error but also uncertainty in the underlying data. Details of the construction of these intervals are given in the SOM.

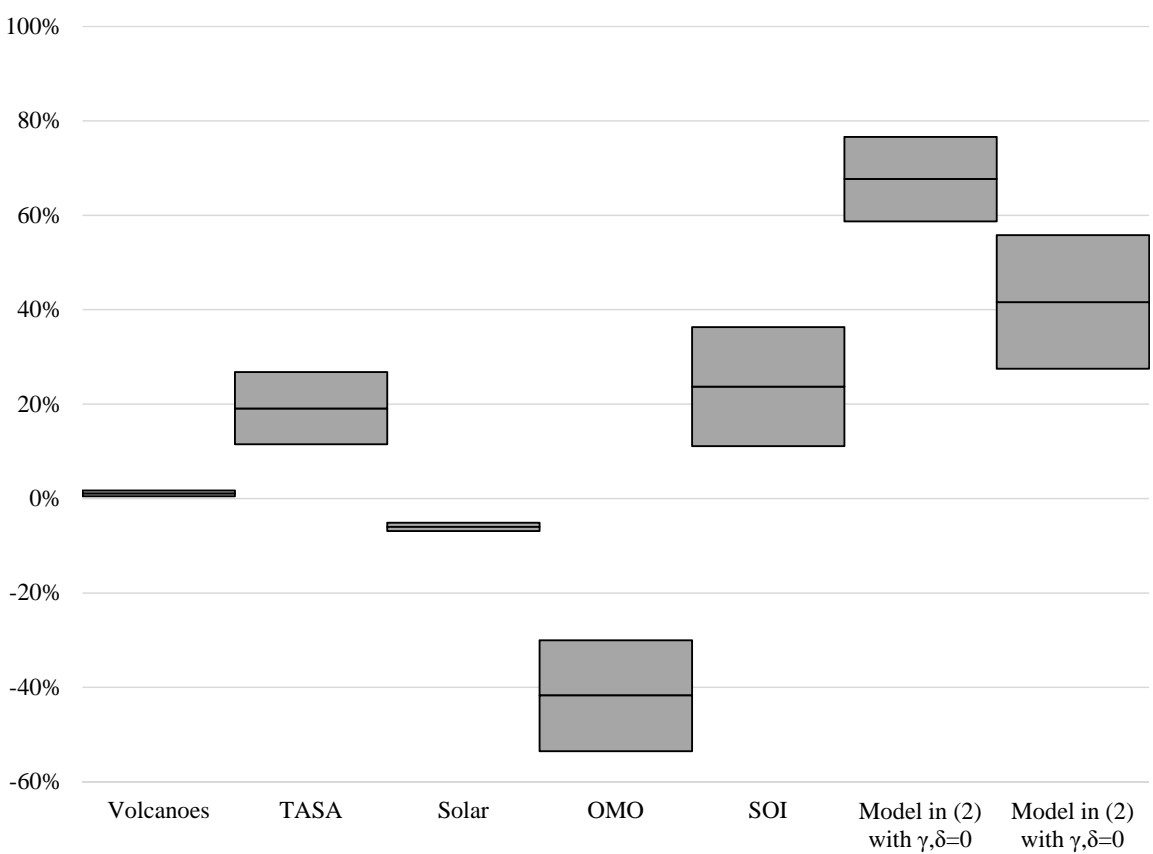

**Figure 2. Estimated Contributions of Key Potential Explanations of the 1998-2013 Episode.** Positive percentages of the missing heat (0.263°C) help to explain it, while negative percentages of the missing heat exacerbate the puzzle. Tropospheric aerosols & surface albedo are abbreviated by TASA. 90% confidence intevals are shown by the two boxes, while the border between the boxes represents the point estimates.

normally and the standard errors consistently for cointegrated temperatures and forcings. A number of previous studies have provided physical and statistical evidence in favor of a cointegrating relationship: Kaufmann *et al.* (2006a, 2006b, 2010, 2013), Pretis (2015), *inter alia*. As explained in the SOM, this procedure also corrects for uncertainty in the forcings data.

Adding only volcanoes to G+Z decreases forcings by $0.036\,\text{W/m}^2$. Predicted GMT decreases by only $0.003°\text{C}\,(0.001, 0.004)°\text{C}$ or about $1.1\%\,(0.5, 1.7)\%$ of (the point estimate of) the missing heat. In the data, forcing from stratospheric aerosols is attributed to volcanic activity, while forcing from tropospheric aerosols is attributed to anthropogenic sulfur dioxide emissions. Vernier *et al.* (2011) refute previous studies that attribute an increase in stratospheric aerosols to emissions. While those authors do not discuss the effect of volcanic activity on the hiatus directly, their Figure 5 suggests that the stratospheric aerosol levels from Mt. Pinatubo subsided until about 1997, and the increase since then has been relatively small.

Similarly, adding only tropospheric aerosols & surface albedo to G+Z decreases forcings by $0.118$ W/m$^2$. Predicted GMT decreases by $0.050°$C $(0.030, 0.070)°$C, or $19.1\%$ $(11.5, 26.8)\%$ of the missing heat. An alternative measure of forcings from tropospheric aerosols based on the sulfur emissions data of Hoesly *et al.* (2018) predicts GMT *increasing* by $0.003°$C $(-0.014, 0.020)°$C, or exacerbating the missing heat by $1.3\%$ $(-5.7, 8.3)\%$. The SOM contains more details of how the alternative data were employed and additional empirical results based on them.

That anthropogenic aerosol emissions appear to explain some of the missing heat using the Hansen *et al.* (2017) data is consistent with the findings of Storelvmo *et al.* (2016) and Kaufmann *et al.* (2011). Unfortunately, although the interval estimate using these tropospheric aerosol data explicitly account for measurement error, it does not cover the point estimate using the data of Hoesly *et al.* (2018), and vice versa. In short, the effect is quite uncertain using either data set.

Solar irradiance *increases* forcings by $0.037$ W/m$^2$, so that the predicted GMT *increases* by $0.016°$C $(0.013, 0.018)°$C, exacerbating the missing heat by $6.0\%$ $(5.1, 6.9)\%$. There is a decline from 2002-2006, but the net effect of solar over 1998-2013 is to increase temperature – not to decrease it. To the extent that solar contributed to the slowdown by decreasing temperatures, the results suggest that solar alone is not sufficient. This finding is not inconsistent with that of Schmidt *et al.* (2014), who examine solar in conjunction with other forcings as an explanation.

The preceding explanations are model forcings, and none of them satisfactorily account for the slowdown either alone or in concert. As previous authors have pointed out, natural variability may play a role, and we now turn to measures of two such types: the OMO and the ENSO.

In order to examine multidecadal oscillations from the OMO as a possible explanation, we let $\gamma \neq 0$, but keep $\delta = 0$. The regressor vector $x_t$ is correlated with the other forcings, and we want to capture the partial effect of the OMO while retaining the total effect of G+Z. In order to do so, we employ the fitted residuals from regressing $x_t$ onto the other forcings as a regressor in the model, rather than using $x_t$ itself. The two approaches – using $x_t$ or its fitted residuals – yield equivalent model fits, but using the fitted residuals fixes the coefficient vector $\alpha$.

The oscillation exacerbates the missing heat by $0.110°$C $(0.079, 0.141)°$C or $41.7\%$ $(30.0, 53.5)\%$. By itself, the fitted OMO worsens the puzzle in the sense that the predicted temperature in 2013 increases to $0.775 + 0.110 = 0.885°$C. The reason for the increase is that the OMO appears to be increasing rather than decreasing during this period. This result contrasts sharply with that of Yao *et al.* (2016), who attribute the hiatus to a much shorter oceanic cycle of 60 years.

The ENSO is quasiperiodic with a period of about 5-6 years. However, roughly every three El Niño episodes are so-called "super El Niños" with much higher amplitudes than the intervening episodes. In other words, the ENSO also oscillates at a decadal scale, roughly 15-18 years. The last two peaks of the longer oscillation were in 1997-98 and 2015-16, coinciding with the beginning and end of the 1998-2013 episode. Letting $\delta \neq 0$ but keeping $\gamma = 0$ yields an increase in the normalized and orthogonalized SOI of 1.162 and thus a decrease of $0.062°$C $(0.029, 0.095)°$C, so that the ENSO explains $23.7\%$ $(11.1, 36.3)\%$ of the missing heat.

All of the explanations so far ignore to some extent that the starting year matters, as has been pointed out by previous authors (Medhaug *et al.*, 2017, e.g.). Not only was 1998 an El Niño year, it was an anomalously warm one. Suppose that the temperature anomaly in 1998 had been equal to that in 1997, $0.389°$C. The same exercise of defining the hiatus using growth rates predicted

by G+Z results in a 2013 temperature anomaly of $0.389 + 0.561 \times 0.426 = 0.628°C$, a decrease of $0.775 - 0.628 = 0.147°C$, which explains $55.9\%$ of the missing heat. In other words, half of the puzzle is explained simply by the construction of the puzzle.

The counterfactual of setting the 1998 temperature to that of 1997 seems effective in explaining the slowdown, but it is extremely *ad hoc*. A similar result is obtained more formally by fitting the model forcings but no natural variation – i.e., by estimating the model in (2) with $\gamma, \delta = 0$. This model predicts the temperature in 1998 to be $0.395°C$ – close to that in 1997 – and increasing to $0.597°C$ by 2013. In other words, a simple model, including all forcings but without the OMO or the ENSO, explains $0.178°C$ $(0.155, 0.202)°C$, or $67.7\%$ $(58.7, 76.6)\%$ – more than half – of the missing heat. Looking at the most comprehensive model with $\delta, \gamma \neq 0$ gives a qualitatively similar result, explaining $41.6\%$ $(27.5, 55.8)\%$.

This finding suggests that the unusually warm year of 1998 – a residual in the model – accounts for most of the apparent slowdown between 1998-2013. It is consistent with the finding of Kosaka and Xie (2013), in the sense that the El Niño year is necessarily followed by La Niña cooling. However, with the results of Pretis *et al.* (2015) and those using the SOI above in mind, we certainly cannot attribute the slowdown to the ENSO uniquely.

Yet there is a new problem given by the high GMTs of $0.760°C$ in 2015 and $0.773°C$ in 2016. The restricted model undershoots these temperatures by more than $0.1°C$. Are these simply outliers, as 1998 was? A natural explanation is the ENSO, because 2015 and 2016 were El Niño years. The model with $\delta \neq 0$ and $\gamma = 0$ – i.e., with SOI but no OMO – undershoots 2015 by more than $0.1°C$ and 2016 by just less than $0.1°C$. In other words, accounting for ENSO does about as poorly as not accounting for ENSO in predicting the temperature in 2015, but improves the prediction for 2016.

Finally, consider the proposed comprehensive model with $\delta, \gamma \neq 0$. The model undershoots 2015 by only $0.042°C$, while overshooting 2016 by less than $0.001°C$ (see Figure 1). We interpret these numbers to mean that the recent high temperatures of 2015 and 2016 are attributable more to the smooth, multidecadal, and somewhat predictable OMO than to the higher-frequency quasi-periodic ENSO. As a result, we can say that 2015 and 2016 were not outliers, and that increases in global mean temperatures may be expected to continue as the OMO continues to put upward pressure on temperatures. Put more simply, the hiatus that appeared to begin in 1998 ended in 2013.

## 2.2 The 2023-2061 Episode

Wyatt and Curry (2014) emphasize that, although evidence supports a secularly varying oscillation like the one that we estimate, future external forcings may alter the amplitude and period of the cycle. Linear detrending may overemphasize this possibility by giving a stochastically trending series with secular oscillations the appearance of a secularly trending series with quasi-periodic or stochastic oscillations. If the long-run trend is indeed anthropogenic, the former is more appropriate than the latter.

We fit a sine function and predict it to 2100, as shown in Figure 3. After crossing zero before 2005, the sine function continues to increase for roughly $76/4 = 19$ years until about 2023, and it then *decreases* for about 38 years until roughly 2061. Figure 3 shows sine functions reflecting a lower and upper $90\%$ confidence interval for the estimated period. This interval is not a prediction interval for a future year, so the plots do not straddle that of the point forecasts. Nor is it constructed from standard errors, which do not reflect correlation of the estimates of the period and phase shift.

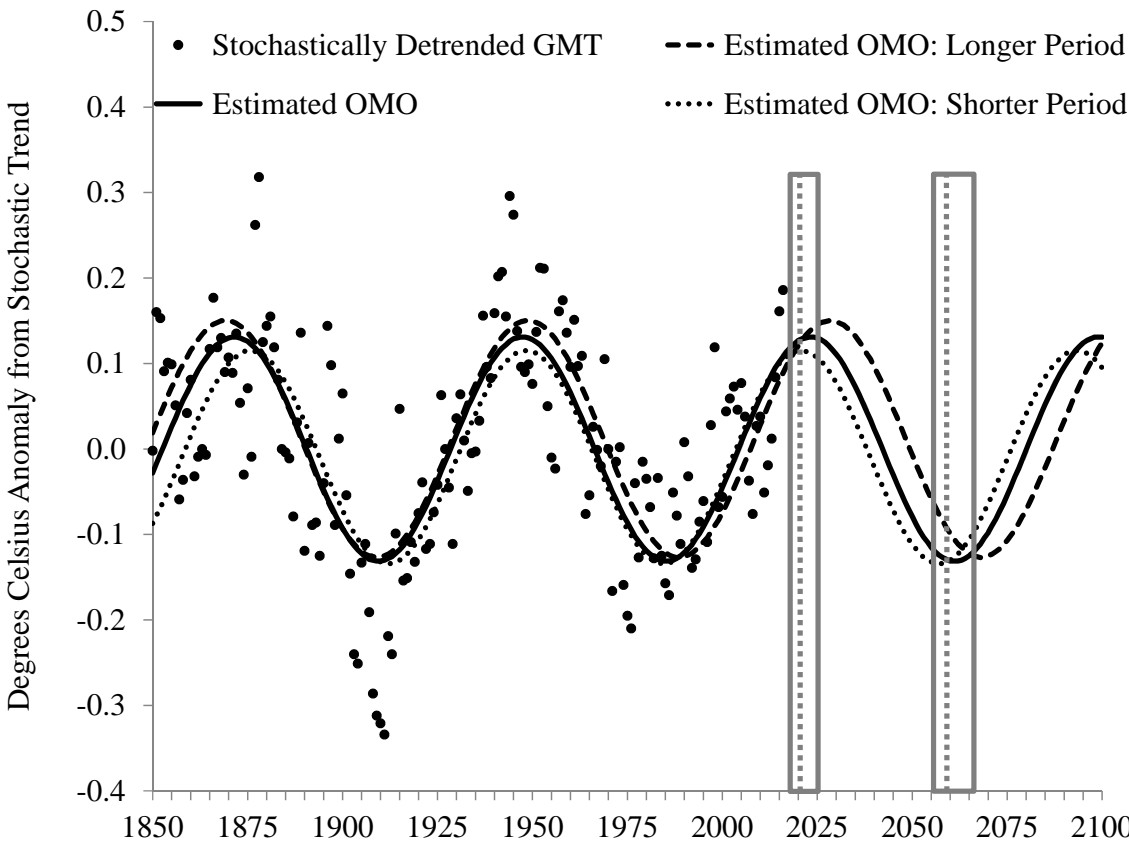

**Figure 3. OMO, 1850-2100.** Estimated and predicted Oceanic Multidecadal Oscillation, with 90% confidence interval on the next downturn and subsequent upturn.

Rather, we rely on an AR(1) bootstrap strategy in the spirit of Poppick *et al.* (2017), which is described in the SOM. The 90% bootstrap confidence interval of the estimated period of 76 years is 73 to 80 years. We date the next peak in the sign function as 2023 – likely falling in the interval 2021-2028 – and the next minimum as 2061 – likely falling in the interval 2057-2068. We note that homogeneous linear detrending results in a period of 72 years, while Trenberth and Shea's (2006) stochastically detrended AMO results in a period of 78 years. Our method, which has the advantages of relating the stochastic trend to forcings by way of a physical model and generates oscillations that are statistically more regular (see the SOM for details), generates an uncertainty interval that plausibly accounts for uncertainty in the stochastic trend.

A downturn in the temperatures due to the ocean cycle implies a slowdown but not necessarily a hiatus in global warming, because the upward trend in forcings may more than offset the downturn. The model in (2) may be used to forecast temperature anomalies conditional on changes in one or more forcings. In our forecasts, we condition on volcanic activity remaining at its 2016 level and the SOI remaining at its temporal mean over 1850-2016. We do not forecast the ENSO, because it is not

periodic enough that long-run forecasts of the ENSO would be very accurate and its estimated effect on temperature is not as large as that from the OMO.

We consider two possible scenarios for non-volcanic forcings, which are closely related to RCP8.5 and RCP6.0. The data in 2016 have already deviated from the RCPs, so we simply match the cumulative growth rates of all forcings $CO_2$ equivalents of all anthropogenic forcings under the two scenarios starting in 2016. RCP8.5 is considered by many to be "business-as-usual," and the average annual growth implied by RCP8.5 is $0.053$ W/m$^2$/yr. Forcings would have to grow at a sustained rate much faster than the recent growth of $0.029$ W/m$^2$/yr over 2013-16 – i.e., since the end of the hiatus and beginning of the recent El Niño period. On the other hand, RCP6.0 has an average annual growth of $0.021$ W/m$^2$/yr, similar to $0.024$ W/m$^2$/yr over the last 76 years – one complete period of the OMO – in order to filter out any multidecadal cyclicality in the forcings themselves.

Two points bear discussion. First, we are ignoring the recalcitrant component of warming (Held *et al.*, 2010), nor are we using a dynamic model to try to capture short-run dynamics. As a result, our model is set up to make conditional forecasts of roughly 5-90 years from the end of the sample. Second, forecasts are conditional on the scenarios mentioned above, but we make no attempt to forecast individual forcings, such as solar or WMGHGs.

Figure 4 shows the sample paths of the conditional forecasts under the two scenarios. Under RCP8.5, anthropogenic forcings increase so much that downturns in the OMO cycle are never again powerful enough to force a hiatus in global warming. The global temperature anomaly increases by $0.022°$C/yr $(0.019, 0.024)°$C/yr on average to nearly $3°$C over the base period by 2100. Nevertheless a slowdown is predicted until about 2061 under RCP8.5, after which point temperatures growth is predicted to accelerate to a much faster rate over multiple decades than that of the historical record. Of course, our forecasts are conditional on ENSO being unrealistically flat. A hiatus could again result from a well-timed super El Niño, such as that in 1997-98.

Under RCP6.0, the temperature increases by about $0.008°$C/yr $(0.006, 0.010)°$C/yr on average. By 2100, temperature anomalies increase to $1.459°$C $(1.292, 1.626)°$C, which is $1.764°$C $(1.598, 1.931)°$C above pre-industrial temperatures, because the base period is $0.305°$C above pre-industrial temperatures, approximated by the 1850-1879 average. While this interval is still below $2°$C, it exceeds the Paris Agreement goal of $1.5°$C.

We see a substantial ebb and flow of the effect of the OMO cycle on temperatures under RCP6.0. Between 2023 and 2061, the dates identified of the next maximum and minimum of the OMO, temperature is predicted to grow by only $0.0001°$C/yr – i.e., virtually no growth. In contrast to the average annual growth of anthropogenic forcings of $0.022$ W/m$^2$/yr under RCP6.0, this projection clearly suggests a future hiatus period that is much longer than the 1998-2013 episode. However, a very crude rule-of-thumb forecast suggests the possibility of a super El Niño in approximately 2034, which could break up the hiatus predicted by the OMO.

The variation in temperatures from the OMO is estimated to be $0.262°$C $(0.249, 0.277)°$C. At its predicted nadir in 2061, temperatures are predicted have increased since 2023 by $0.100°$C $(0.098, 0.102)°$C under RCP6.0 or $0.486°$C $(0.479, 0.494)°$C under RCP8.5, meaning that they would have increased by $0.262 + 0.100 = 0.362°$C under RCP6.0 or $0.262 + 0.486 = 0.748°$C under RCP8.5 without the OMO. Based on the point estimates, we expect the variation of the OMO to mask the underlying warming trend by $1 - 0.486/0.748 = 35\%$ under RCP8.5 and $1 - 0.100/0.362 = 72\%$ under RCP6.0.

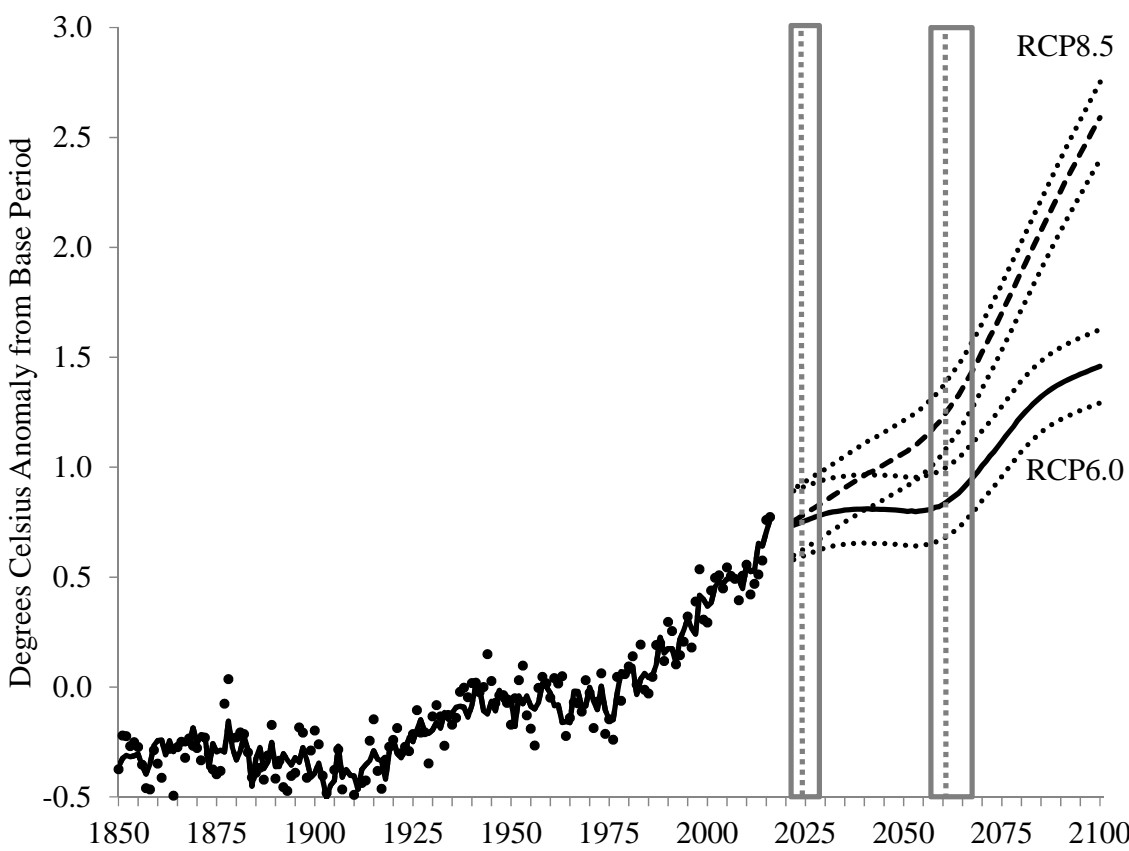

**Figure 4. Conditional Forecasts of Temperature Anomalies, 2022-2100.** Until 2016, dots represent actual temperature anomaly data and the solid line represents the temperature anomaly predicted by the model. Scenario labeled RCP8.5 uses RCP8.5 growth rates for anthropogenic forcings from a starting point of 2016, and similar for RCP6.0. 90% confidence bands are shown.

## 3    Summary and Implications for Policy Evaluation

It is no exaggeration to say that the 1998-2013 apparent hiatus in the otherwise evident trend of warming global mean temperatures has generated controversy. From a scientific point of view, a number of researchers have put forth differing explanations backed up by plausible physical models joined with sound statistical methods. Because of the critical importance of climate
5    change to human systems – economic, political, etc. – the controversy has spilled over into the arena of public and political debate, where the lack of warming is viewed as empirical validation by those skeptical of global warming. Lack of consensus about the cause only adds to such doubt.

In this paper, we disentangle some of the causes of the 1998-2013 hiatus and subsequent run-up in temperatures using a modern statistical technique, a semiparametric cointegrating regression, based on an energy balance model. Our main findings
10    for this period suggest that the three main factors driving the hiatus were (a) the unusually warm year of 1998, even conditional

on the ENSO, (b) the ENSO itself, and (c) the increase in tropospheric aerosols during that period, though the latter is measured with a high degree of uncertainty. Other potential causes that we investigate had considerably less impact or else an accelerating rather than confounding impact on rising temperatures. Our statistical model not only explains much of the hiatus but also explains the rapid warming since 2013. We find that this warming marks the end of the hiatus, in contrast to some findings in

the literature (Chen and Tung, 2014, Knutson *et al.*, 2016, e.g.) but consistently with that of Sévellec and Drijfhout (2018)

Further, fitting the mean of the distribution of detrended ocean temperature anomalies (an oceanic multidecadal oscillation) to a periodic function enables us to make forecasts of the global mean temperature conditional on forcing scenarios. If forcings grow at the same rate as they have for the past 76 years, the estimated period of the OMO, we can expect a longer hiatus in global warming from about 2023 to about 2061, roughly 3-4 decades. The controversy of the recent 15-year hiatus is a

precursor to that which may result from a much longer one. Kaufmann *et al.* (2017) recently showed a correlation between climate skepticism and locally cooler (or less warm) temperatures in the US. If the lack of warming indeed drives doubt, three decades of no warming is indeed a long period to fuel skepticism. Nevertheless, on the current trajectory, we can expect the decades following the next hiatus to push well past the $1.5°C$ goal of the Paris Agreement and even past $2°C$.

Even though our model makes use of spatially disaggregated sea surface temperatures, our results have nothing to say directly

about regional differences in temperature oscillations. As emphasized by Kaufmann *et al.* (2017) and many other authors, the effects are spatially heterogeneous. Based on estimated warming trends displayed in Figure S.1 of the SOM, we speculate that the effect of a future hiatus will be more noticeable in the vicinity of the Pacific and Indian Oceans than in the vicinity of the Atlantic Ocean, because the latter has more strongly increasing trends.

It may be useful to assign a probability to the possibility of a future multidecadal hiatus. Such a forecast would require

more information and entail more uncertainty than the conditional forecasts above, because a probability distribution would be needed for future forcings. Rather than try to forecast forcings, one could base such a forecast on, say, expert opinion of the likelihood of forcing scenarios. Suppose, for example, that one believes that forcings will increase at an average rate of $w$ per year, where $w$ is a random variable symmetrically distributed around RCP6.0. Figure 4 suggests that, roughly speaking, scenarios with weaker growth will result in a future hiatus, while those with stronger growth will not. Ignoring the uncertainty

associated with the conditional forecasts, one with such a prior could make a prediction that a multidecadal hiatus will occur with a probability of roughly $50\%$. We emphasize, however, the inherent uncertainty in such an exercise, even taking into account our allowance for uncertainty in the data and estimates.

Our forecasts are *conditional* on hypothetical concentration pathways. We cannot and do not suggest that policy should be based on our results. Rather, we seek to inform scientists and policymakers of the possibility of a warming hiatus due to a

natural cycle. Such a cycle may be expected to have a confounding effect on policy evaluation, because a natural downturn may be mistaken for the effectiveness of mitigation. Quasi-experimental statistical evaluation of such policies must take into account this effect to avoid mistaking a failed policy for a successful one.

*Code and data availability.*  Data, code, and appendices are available as Supplementary Online Material.

*Competing interests.* The authors declare no competing interests.

*Acknowledgements.* The authors appreciate useful feedback from Buz Brock, Neil Ericsson, David Hendry, Luke Jackson, and participants of the 2017 Conference on Econometric Models of Climate Change (Nuffield College, University of Oxford), a seminar at the Korea Energy Economics Institute, and a colloquium at the University of Missouri. All errors are our own.

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
