# Peer review of "Dating hiatuses: a statistical model of the recent slowdown in global warming – and the next one"

_Earth System Dynamics, 2018_

## Referee Comment (RC1) · Anonymous Referee #1 · 3 Jan 2019

This paper focuses on the detection and attribution of the temperature hiatus over the last decade, as a hot issue for the climate change studies. Based on the semi-parametric cointegrating regression approach, the authors give one explanation of the temperature hiatus by considering many physical causes, which is useful for improving the understanding. However, there are two questions I concerned. One is the influence of data quality on the results, and the other is the influence of the temperature hiatus on the whole temperature variability in the future. Some specific comments are given as follows: (1) The volcanoes of course influence the temperature variability, with a contribution of 1% explained by the authors. However, as is known, the results from regressive approaches have big uncertainty, and so the 1% contribution is real or just

bias? (2) The quality and quantity of observations prior to the satellite era are questionable. How much actual observation data is included prior to 1970s in the monthly HadSST3 used in this study? Please provide that information for the credibility of the results, especially for the results in Figure 2. Besides, if the results have biases only using the HadCRU data? as this data set has biases at monthly scales. (3) I cannot understand why using the ENSO to explain the temperature Hiatus at decadal scales, because it mainly exhibits oscillatory variations at interannual scales. Further, the authors also investigate the temperature Hiatus in the future based on the OMO, but not the ENSO. How to coordinate the influence of the OMO and ENSO on the temperature variability? (4) The authors discuss the temperature Hiatus in the future using the Sin extrapolation of OMO. As mentioned above, is there any uncertainty for the practice? Moreover, the contribution of the temperature Hiatus (that is, the oscillatory variations of temperature) to the whole temperature variability (especially more significant increase) in the future should be clarified, as it is more important for the policy-making, as discussed in Conclusions. Besides, the regional difference about the results can be simply discussed.

---

## Referee Comment (RC2) · Anonymous Referee #2 · 10 May 2019

Global warming has long attracted the attention of the climate research community, but also socioeconomic fields, for its expected huge impacts on the Earth's climate and our living environments. To date, we have not yet sufficiently understood the physical mechanisms accounting for the causality between warming and anthropogenic and natural processes. Although the existing studies using numerical models have provided important information for understanding global warming and climate change, the known inadequacy, uncertainties, and biases in models make us today still not clearly understand warming mechanisms by model alone. This study used a semiparametric statistical regression model, and proposed a new oceanic multidecadal oscillation index measuring the multibasin contribution to global mean temperature, to date and attribute

the temperature hiatus from the perspective of physical processes and statistical features. The approaches and results are helpful to further our knowledge of the warming temperature oscillations and climate change. As is known, the biggest disagreement with hiatus comes from data uncertainty. So the usefulness of this study lies in the worthy addition to our study approaches and thinking perspectives for global warming. Regarding the projections, the credibility is not enough to support our policy-making, instead add risks thereof. So the suggestion is that authors limit the implications of this study in the range of study methodology and perspective, and include a caveat of uncertainty in the projection into the conclusion section.

---

## Author Comment (AC1) · 5 Jun 2019

Referee Comment: This paper focuses on the detection and attribution of the temperature hiatus over the last decade, as a hot issue for the climate change studies. Based on the semiparametric cointegrating regression approach, the authors give one explanation of the temperature hiatus by considering many physical causes, which is useful for improving the understanding. However, there are two questions I concerned. One is the influence of data quality on the results, and the other is the influence of the temperature hiatus on the whole temperature variability in the future.

[Figure]

Author Response: Thank you for your careful reading of our manuscript and suggestions for improvement. We carefully considered the issues you raised and revised the manuscript accordingly. Our responses to each point raised ensue.

Referee Comment: Some specific comments are given as follows: (1) The volcanoes of course influence the temperature variability, with a contribution of 1% explained by the authors. However, as is known, the results from regressive approaches have big uncertainty, and so the 1% contribution is real or just bias?

Author Response: We spent much time thinking about uncertainty when we conducted our analysis. The $1\%$ for volcanoes is just a point estimate, and our interval estimate of $(0.3, 1.9)\%$ suggests that the real contribution could be higher or closer to zero. This range comes from not only the "usual" uncertainty of the regression error, but also uncertainty in the underlying data, which we tried to accommodate in a reasonable but admittedly *ad hoc* manner based results in the IPCC chapter of Myrhe *et al.* (2013), explained in our SOM. We use the commonly employed data of Hansen *et al.* (2017), in which forcing from stratospheric aerosols is attributed to volcanic activity, while forcing from tropospheric aerosols is attributed to anthropogenic sulfur dioxide emissions. Vernier *et al.* (2011) refute previous studies that attributed an increase in stratospheric aerosols to emissions, which gives us some confidence that we are interpreting these measurements appropriately. While those authors do not discuss the effect of volcanic activity on the hiatus directly, their Figure 5 suggests that the stratospheric aerosol levels from Mt. Pinatubo subsided until about 1997, and the increase since then – to which we attribute $(0.3, 1.9)\%$ of the recent hiatus – has been relatively small.

Author Action: **Amended** footnote 6 to explain that uncertainty results from both statistical error and data. **Added** some of the explanation above to page 6 (lines 4-8) of the revised manuscript.
Referee Comment: (2) The quality and quantity of observations prior to the satellite era are questionable. How much actual observation data is included prior to 1970s in the monthly HadSST3 used in this study? Please provide that information for the credibility of the results, especially for the results in Figure 2. Besides, if the results have biases only using the HadCRU data? as this data set has biases at monthly scales.

Author Response: We certainly agree. Keeping in mind that the HadSST3 data already aggregate observations within five degrees latitude and longitude in order to alleviate some of the observational uncertainty, the maximum number of observations possible is $36 \times 72 = 2,592$ per month and $2,592 \times 12 = 31,104$ per year. We include a figure here (please see below) and in the revised version of the SOM that shows the actual number of observations per year. Observations are based on readings from buoys and ships. A maximum of $17,391$ is attained in 1979 and the number is slightly lower but steady since then, as the referee points out. Prior to the 1970's the number of observations generally increases over time, but with noticeable dips during major international disruptions, such as World War II, World War I, and the American Civil War. Even at its lowest in 1866, just after the American Civil War, the number of observations $(2,414)$ still exceeds a thousand.

We use temperature data on both sides of the model. On the left-hand side, we use the global mean temperature anomaly from HadCRUT4, which also includes land. Static bias and idiosyncratic/short-run error from uncertainty in the measurement of the data are picked up by the intercept $\alpha_0$ and regression error $\eta_t$ respectively in equation (2) of the paper. However, a change in quality and quantity of temperature measurements over time may cause heteroskedasticity of unknown form in the regression error. Our coefficient estimates would be less precise as a result, but should still be statistically consistent.

On the right-hand side, we use disaggregated HadSST3 data. By design, our method for smoothing the cyclical component should eliminate any idiosyncratic/short-run un-

certainty in these data. If there is a static bias throughout the time span, it is picked up by a non-zero estimate of $\theta_4$ in equation (S.1) of the SOM, which we estimate to be nearly zero (Table S.1). Since we use $\theta_4$ to build the cyclical component, any bias in our estimate of $\theta_4$ is picked by the intercept $\alpha_0$ in equations (1) and (2), but should not affect the other coefficients estimates used to make our inferences. However, if the bias changes over time in a non-idiosyncratic way, it would be a more complicated problem to explicitly model, and we leave it for future consideration.

Author Action: **Included** new figure and **added** exposition similar to that above regarding the number of observations on page 3 (lines 6-10) of the SOM. **Added** exposition similar to that above regarding bias on page 6 (line 12) through page 7 (line 3) of the SOM.

Referee Comment: (3) I cannot understand why using the ENSO to explain the temperature Hiatus at decadal scales, because it mainly exhibits oscillatory variations at interannual scales. Further, the authors also investigate the temperature Hiatus in the future based on the OMO, but not the ENSO. How to coordinate the influence of the OMO and ENSO on the temperature variability?

Author Response: As the referee points out, the ENSO oscillates at an interannual scale. It is quasiperiodic with a period of about 5-6 years. Roughly every three El Niño episodes are so-called "super El Niños" with much higher amplitudes than the intervening episodes. In other words, the ENSO also oscillates at a decadal scale, roughly 15-18 years. The last two peaks of the longer oscillation were in 1997-98 and 2015-16, marking the beginning and the end – we claim – of the recent hiatus. Essentially, our model suggests that cooling after 1997-98 offset and temporarily masked warming from anthropogenic and other causes until 2015-16.

Looking forward, the referee is correct that we did not predict the ENSO and therefore conditioned our temperature forecasts on scenarios with no ENSO. The ENSO is not

periodic enough that long-run forecasts of the ENSO would be very accurate and its estimated effect on temperature is not as large as that from the OMO. Put differently, we believe that the loss in forecast accuracy from conditioning our temperature forecasts on those of the ENSO would exceed the benefit from doing so. However, as a very crude forecast, we may expect a super El Niño again in about 2034, which could break up the hiatus predicted by the OMO.

Author Action: **Added** exposition similar to that above on the decadal scale to page 6 (lines 32-35). **Added** exposition similar to that above on prediction to page 9 (lines 1-2, 20-22, and 30-32).

Referee Comment: (4) The authors discuss the temperature Hiatus in the future using the Sin extrapolation of OMO. As mentioned above, is there any uncertainty for the practice? Moreover, the contribution of the temperature Hiatus (that is, the oscillatory variations of temperature) to the whole temperature variability (especially more significant increase) in the future should be clarified, as it is more important for the policy-making, as discussed in Conclusions. Besides, the regional difference about the results can be simply discussed.

Author Response: Every forecast comes with uncertainty. We have taken into account uncertainty in the historical data and uncertainty in the parameter estimates in generating our interval forecasts for the start and end of the next hiatus. We explain our procedure in the SOM.

We estimate the variation in temperature from the OMO to be $0.26°C$ $(0.25, 0.28)°C$. At its predicted nadir in 2061, temperatures are predicted have increased since 2023 by $0.11°C$ $(0.11, 0.11)°C$ under RCP6.0 or $0.50°C$ $(0.49, 0.51)°C$ under RCP8.5, meaning that they would have increased by $0.37°C$ under RCP6.0 or $0.76°C$ under RCP8.5 without the OMO. Based on the point estimates, we expect the variation of the OMO to

mask the underlying warming trend by $34\%$ under RCP8.5 and $70\%$ under RCP6.0. It is this $70\%$ that we expect to result in an apparent hiatus over this period.

Even though our model makes use of spatially disaggregated sea surface temperatures, our results have nothing to say directly about regional differences in the effects of the oscillations. The referee is certainly correct that the regional differences are important for policymakers, so it seems appropriate to speculate on these effects, which we do in the revision.

Author Action: **Added** exposition similar to that above on variations to page 9 (line 33) through page 10 (line 2). **Added** exposition speculating on regional differences to page 11 (lines 19-24).

[Figure]

Fig. 1.

---

## Author Comment (AC2) · 5 Jun 2019

Referee Comment: Global warming has long attracted the attention of the climate research community, but also socioeconomic fields, for its expected huge impacts on the Earth's climate and our living environments. To date, we have not yet sufficiently understood the physical mechanisms accounting for the causality between warming and anthropogenic and natural processes. Although the existing studies using numerical models have provided important information for understanding global warming and climate change, the known inadequacy, uncertainties, and biases in models make us today still not clearly understand warming mechanisms by model alone. This study

used a semiparametric statistical regression model, and proposed a new oceanic multidecadal oscillation index measuring the multibasin contribution to global mean temperature, to date and attribute the temperature hiatus from the perspective of physical processes and statistical features. The approaches and results are helpful to further our knowledge of the warming temperature oscillations and climate change. As is known, the biggest disagreement with hiatus comes from data uncertainty. So the usefulness of this study lies in the worthy addition to our study approaches and thinking perspectives for global warming. Regarding the projections, the credibility is not enough to support our policy-making, instead add risks thereof. So the suggestion is that authors limit the implications of this study in the range of study methodology and perspective, and include a caveat of uncertainty in the projection into the conclusion section.

Author Response: Thank you for your careful reading of the manuscript and suggestions for improvement. We have done our best to take into account unceraint in the underlying data and in estimation of the model parameters, but projections are of course necessarily uncertain and we now include such a caveat. In the revision, we now emphasize a usefulness in the *evaluation* of policy: the natural cycle that we estimate can have a confounidng influence on mitigation, so our results may be taken as a warning to those conducting quasi-experimental evaluations of such policies.

Author Actions: **Clarified** the use to inform policy throughout. **Added** to the conclusion: "We emphasize, however, the inherent uncertainty in such an exercise, even taking into account our allowance for uncertainty in the data and estimates." **Added** to the conclusions: "Our forecasts are *conditional* on hypothetical concentration pathways. We cannot and do not suggest that policy should be based on our results. Rather, we seek to inform scientists and policymakers of the possibility of a warming hiatus due to a natural cycle. Such a cycle may be expected to have a confounding effect on policy evaluation, becase a natural downturn may be mistaken for the effectiveness

of mitigation. Quasi-experimental statistical evaluation of such policies must take into account this effect to avoid mistaking a failed policy for a successful one."

―――――――――――――――――

---

## Referee Report (RR1)

**Review: Dating Hiatuses: A statistical model of the recent slowdown in global warming – and the next one (esd-2018-81)**

This paper uses econometric techniques to investigate the causes for a hiatus in warming and forecast whether another hiatus will occur in the future. The main results indicate that three factors drove the hiatus, the unusually warm year of 1998, the ENSO itself, and increases in tropospheric aerosols. With regard to the future, the authors conclude that the rate at which forcings rise will influence whether there is a hiatus in the future. These results (and to some degree the econometric results that are used to generate them) will interest the readers of *Earth System Dynamics.* As such, the manuscript should be considered for publication. The current form of the manuscript is close to being ready for publication. As described below, I have two substantive concerns and some minor concerns about the presentation/interpretation of results.

*Substantive Comments*

My first substantive concern focuses on the Oceanic Multidecadal Oscillation (OMO) and the empirical methods used to identify its effects. The OMO phenomenon needs to be described in more detail. Specifically, the authors need to define the OMO explicitly, briefly review the physical mechanisms thought to drive it, and how the OMO affects surface temperature.

The authors also need to investigate the degree to which their results are sensitive to the methods used to estimate its effect on temperature. On page 8, line 1, the authors state that the OMO is estimated using a sine function. The authors should explain why a sine function is used. Figure 1 suggests that the sine function is used to fit a time series for stochastically detrended GMT. But the authors do not explicitly define the data used to fit the sine function nor do the authors describe how they stochastically detrend global mean temperature. Because many readers are not econometricians, the authors need to show the equation(s) that they use to stochastically detrend global mean temperature.

Finally, the authors need to investigate the degree to which this approach affects their results. My cursory review of the literature indicates that one set of authors calculate an index for the Atlantic Multidecadal Oscillation index from linearly detrended North Atlantic sea surface temperature anomalies while others identify the OMO signal using empirical orthogonal functions. How would the results reported in this manuscript change if they used one of these methods to estimate the effects of the OMO, instead of using the stochastically detrended global mean temperature?

My second substantive concern focuses on the time series used to represent the cooling effects of tropospheric sulfates. In Figure 1, the authors identify the ability of various forcings to account for the missing heat, as represented by degrees Celsius anomaly from the base period. This is a very straightforward and understandable way to approach the problem. My issue here concerns the forcings used to simulate the model, which are the time series used to simulate the GISS model. In general, the forcings used to simulate the GISS model are highly stylized. They are largely linear with little variation in growth rates over time. This is especially true for reflective tropospheric aerosols. This linearity is different from the time series for anthropogenic sulfur emissions that are assembled by Steven J. Smith (and others) at the Pacific Northwest National Laboratory. These data are updated such that it is compatible with the sample period used by the authors

(https://www.geosci-model-dev.net/11/369/2018/gmd-11-369-2018.pdf). I suggest that the authors investigate the degree to which their results are sensitive to the forcing used by redoing their analysis with the time series from the paper by Hoesly *et al*., (2017).

*Minor Comments*

**Page 3**: "explain past multidecadal cooling or hiatus periods, such as the decades following the temperature spikes in about 1877 and 1943." On page 271 Kaufmann et al (2006) (cited by the authors) write "The radiative forcing of anthropogenic sulfur emissions increases at about the same rate as greenhouse gases between 1944 and 1976. As a result, there is relatively little net increase/decrease in total radiative forcing and therefore, global surface temperature. The timing of these temperature effects is consistent with results obtained from model simulations (Andronova and Schlesinger, 2000; Tett et al., 1999)." The authors should reconcile their statement about the hiatus with the explanation based on a slight decrease in total forcing.

**Page 5** Lines 1-3 describe the method used to covert a change in W/m2 to temperature ($0.536 + 0.561 \times 0.431 \approx 0.777^{o}C$). The authors should explain where 0.430 'comes from.'

**Page 11** "Our main findings for this period suggest that the three main factors driving the hiatus were (a) the unusually warm year of 1998, even conditional on the ENSO, (b) the ENSO itself, and (c) the increase in tropospheric aerosols during that period, though the latter is measured with a high degree of uncertainty" This is a very important component of the authors results, but these results are not really clear in the abstract. The authors should edit this abstract to make these results clearer. Also the abstract should highlight the result that the occurrence of a future hiatus depends on in part on the rate at which forcing grows.

**Lieteratiure cited**

Hoesly, R. M., Smith, S. J., Feng, L., Klimont, Z., Janssens-Maenhout, G., Pitkanen, T., Seibert, J. J., Vu, L., Andres, R. J., Bolt, R. M., Bond, T. C., Dawidowski, L., Kholod, N., Kurokawa, J.-I., Li, M., Liu, L., Lu, Z., Moura, M. C. P., O'Rourke, P. R., and Zhang, Q.: Historical (1750–2014) anthropogenic emissions of reactive gases and aerosols from the Community Emission Data System (CEDS), Geosci. Model Dev. Discuss., doi:10.5194/gmd-2017-43, in review, 2017.

---

## Referee Report (RR2)

Review: Dating Hiatuses: A statistical model of the recent slowdown in global warming -- and the next one.

I thank the authors for their careful attention to my comments. They are addressed satisfactorily in the revised manuscript. Indeed, their manuscript successfully marries sophisticated statistical analysis with a sophisticated understanding of the science. As such, I believe the manuscript should be published.

I do have one last suggestion for the final manuscript. On pages 6-8, the authors repeatedly refer to the absolute and percentage of the missing heat that is associated with a given variable. For example, on page 6 (lines 9 – 10) they argue that volcanoes account for about 1.1 % of the missing heat and again on page 7 (lines 34-35) "that ENSO explains 23.7% of the missing heat.

I suggest that the authors create a figure that allows readers to see these absolute amounts or percentages in one place. This will allow reader to compare various contributions. I realize that many of these findings are implicit in Figure 1. But the slopes of the temperature lines are much harder to quantify and interpret relative to a bar chart that explicitly shows the quantity of missing heat that is associated with volcanoes, ENSO, etc.

Also figure 1 does not show the change in temperature previous studies attribute to various factors. Indeed, the bar chart could also show the quantity of 'missing heat' that previous analyses attribute to a particular variable, such as volcanoes or ENSO. When possible, this would help readers compare the authors' results to those generated by previous studies, which the authors review on pages 1-2.

---

## Author Response (AR2)

**Authors' Response to:** Review: Dating Hiatuses: A statistical model of the recent slowdown in global warming – and the next one (esd-2018-81)

This paper uses econometric techniques to investigate the causes for a hiatus in warming and forecast whether another hiatus will occur in the future. The main results indicate that three factors drove the hiatus, the unusually warm year of 1998, the ENSO itself, and increases in tropospheric aerosols. With regard to the future, the authors conclude that the rate at which forcings rise will influence whether there is a hiatus in the future. These results (and to some degree the econometric results that are used to generate them) will interest the readers of *Earth System Dynamics*. As such, the manuscript should be considered for publication. The current form of the manuscript is close to being ready for publication. As described below, I have two substantive concerns and some minor concerns about the presentation/interpretation of results.

Thank you for your careful reading of our work and constructive comments, which have improved the manuscript! Please see our detailed responses below.

*Substantive Comments*
1. My first substantive concern focuses on the Oceanic Multidecadal Oscillation (OMO) and the empirical methods used to identify its effects.
   a. The OMO phenomenon needs to be described in more detail. Specifically, the authors need to define the OMO explicitly, briefly review the physical mechanisms thought to drive it, and how the OMO affects surface temperature.
   - **Added** a paragraph to the introduction: "Keenlyside et al. (2008) note an acceptance in the literature that the AMO and multidecadal periodic global temperature fluctuations more generally are related to the AMOC. Drijfhout et al. (2014) posit a physical link through changes in heat uptake across multiple oceans. Specifically, a weakening/strengthening of AMOC, meaning less/more convection and turbulent heat loss, leads to increased/decreased net heat uptake and therefore lower/higher surface temperatures. The change in net heat uptake occurs across multiple ocean basins, so we label the resulting multidecadal temperature cycle the Oceanic Multidecadal Oscillation (OMO). In other words, the OMO is the global sea surface temperature fluctuation resulting from changes in the thermohaline circulation. The OMO contrasts with the AMO in scope -- the latter is defined for the North Atlantic -- but the two may be highly correlated with similar periodicities."
   b. The authors also need to investigate the degree to which their results are sensitive to the methods used to estimate its effect on temperature.
   - **Added a paragraph and table** with additional investigations to the SOM: "The estimate of the aggregate effect of the OMO (net of forcings) reported in the paper for the optimal order, $(p_1,q_1)=(2,0)$, is 0.11°C (0.08,0.14)°C over 1998-2013, or a 41.7% (30.0,53.5)% exacerbation of the puzzle of the missing heat. As a robustness check on the effect of selection of $p_1$ and $q_1$, we varied each ±1 from the optimal order: (1,0), (3,0), (2,1), yielding estimates of the aggregate effects of 0.122°C (47.6% exacerbation), 0.092°C (35.9% exacerbation), and 0.029°C (11.3% exacerbation). The estimates with (1,0) or (2,0) seem the most plausible, because the additional terms of models with (3,0) and (2,1) generate estimates with very large magnitudes and opposing signs, a classic sign of near multicollinearity."

c. On page 8, line 1, the authors state that the OMO is estimated using a sine function. The authors should explain why a sine function is used.

- **Added several paragraphs** to the empirical results section: "The SOM (Supplementary Online Material) contains a detailed description of the methodology used to estimate the distribution of the OMO, the probability density function of which we denote by f(r) with support $[r^-, r^+]$ (see the bottom panel of Figure S.3). Our methodology omits both long-run temporal temperature trends to avoid cointegration with h_{t} and idiosyncratic noise to avoid over-fitting very short-run fluctuations in GMT using sea-surface temperatures.

  Our method is intuitively similar to band-pass filters used to identify cycles in economic and other oscillating time series, but it is not executed in the frequency domain. Consistent with an energy balance model, filtering long-run stochastic trends (low frequency on the spectrum) is accomplished by regressing out the anthropogenic signal (see Figure S.1 of the SOM). In contrast, linear detrending (Enfield *et al.*, 2001) unnecessarily assumes a constant growth rate of the anthropogenic signal. More modern approaches, such as that of Trenberth and Shea (2006) and Lenton *et al.* (2017) also filter the anthropogenic signal, but they do so indirectly as a difference between two temperature series both subjected to the same stochastic trend.

  Detrending results in a noisy oscillation. High-frequency filtering could be accomplished using spectral methods if the time series were long enough or if the noise were "quiet" enough. However, it is difficult to identify a cycle with a 65-80 year period from a time series of 167 years in the presence of high-frequency cycles with substantial amplitudes, such as ENSO and the solar cycle. By focusing on a single periodic function, we narrow the desired frequency band enough to identify the cycle and to estimate it very transparently in the time domain. Specifically, we fit the stochastically detrended temperatures to a single periodic function (see Table S.1 and Figure S.4 of the SOM)."

d. Figure 1 suggests that the sine function is used to fit a time series for stochastically detrended GMT. But the authors do not explicitly define the data used to fit the sine function nor do the authors describe how they stochastically detrend global mean temperature. Because many readers are not econometricians, the authors need to show the equation(s) that they use to stochastically detrend global mean temperature.

- **Amended.** The description of our method for constructing the OMO is mostly contained in the SOM for the sake of brevity. We are concerned that too many technical details may be off-putting to some readers, but we wholeheartedly agree with the referee that the SOM should contain these details. We have now revised this part of the SOM, and the methodology should now be much clearer. If the explanation is still unclear, we can try again. (Or, if the editor and referee would prefer, we can bring these out of the SOM and into the paper at the expense of lengthening it.)

e.  Finally, the authors need to investigate the degree to which this approach affects their results. My cursory review of the literature indicates that one set of authors calculate an index for the Atlantic Multidecadal Oscillation index from linearly detrended North Atlantic sea surface temperature anomalies while others identify the OMO signal using empirical orthogonal functions. How would the results reported in this manuscript change if they used one of these methods to estimate the effects of the OMO, instead of using the stochastically detrended global mean temperature?

- **Added $R^2$'s for nonlinear regressions and relevant discussion** to the SOM: "The table also shows $R^2$'s from the three regressions. These statistics should be interpreted with caution, both because they are $R^2$'s from nonlinear regressions and because the regressions have the same functional form of time but different regressands. Nevertheless, we interpret them as evidence that the periodic function is a better approximation to the oscillation estimated using the proposed detrending method."

- **Added discussion** to the manuscript: "We note that homogeneous linear detrending results in a period of 72 years, while Trenberth and Shea's (2006) stochastically detrended AMO results in a period of 78 years. Our method, which has the advantages of relating the stochastic trend to forcings by way of a physical model and generates oscillations that are statistically more regular (see the SOM for details), generates an uncertainty interval that plausibly accounts for uncertainty in the stochastic trend."

- As we reported in the SOM, we tried alternative methods of detrending from the literature, including linear detrending and subtracting out alternative measurements of the stochastic temperature trend along the lines of Trenberth and Shea (2006). We hope that this is now clearer.

2. My second substantive concern focuses on the time series used to represent the cooling effects of tropospheric sulfates. In Figure 1, the authors identify the ability of various forcings to account for the missing heat, as represented by degrees Celsius anomaly from the base period. This is a very straightforward and understandable way to approach the problem. My issue here concerns the forcings used to simulate the model, which are the time series used to simulate the GISS model. In general, the forcings used to simulate the GISS model are highly stylized. They are largely linear with little variation in growth rates over time. This is especially true for reflective tropospheric aerosols. This linearity is different from the time series for anthropogenic sulfur emissions that are assembled by Steven J. Smith (and others) at the Pacific Northwest National Laboratory. These data are updated such that it is compatible with the sample period used by the authors (https://www.geosci-model-dev.net/11/369/2018/gmd-11-369-2018.pdf). I suggest that the authors investigate the degree to which their results are sensitive to the forcing used by redoing their analysis with the time series from the paper by Hoesly *et al.*, (2017).

- **New analysis and section added to the SOM.** We acknowledge this important point and we really wrestled with how to address this comment. Another important difference between the Hansen data and the Hoesly data is that the former are expressed in $W/m^2$ and based on concentrations, while the latter are emissions data. Additional models – of the carbon cycle, e.g. – are required to convert emissions into forcings. Some authors, such as Kaufmann *et al.* (2006), have done this, but we are reluctant to add the additional models that would be required for most forcings, because (a) such an approach could just as easily be criticized, (b) estimation of the model contributes another layer of uncertainty to the data, and (c) most of the forcings should be measured without very much uncertainty, per Myhre *et al.* (2013), and we are already taking this uncertainty into account.

  Of course, tropospheric sulfates are exceptional, both because of the large uncertainty in the data and in their short residence times in the atmosphere, and we interpret the referee's concern to be mainly about measuring these. Because of the short residence time, it seems reasonable to equate concentrations with emissions within the time scale of a one-year increment. Please see the section we added to the SOM for additional details.

  The data sources *do* make a difference, and we now report this difference in the paper. How can we interpret this discrepancy at the end of the day? Clearly, Myhre *et al.* (2013) are right that there is much uncertainty associated with tropospheric aerosols! Our results included a very wide uncertainty interval around this effect, and the additional analysis supports perhaps an even wider uncertainty interval.

- We also **made minor revisions** to our input data and measures of uncertainty following a close comparison of the Hoesly and Hansen data. The numbers throughout the paper have changed a bit as a result.

*Minor Comments*

1. Page 3: "explain past multidecadal cooling or hiatus periods, such as the decades following the temperature spikes in about 1877 and 1943." On page 271 Kaufmann et al (2006) (cited by the authors) write "The radiative forcing of anthropogenic sulfur emissions increases at about the same rate as greenhouse gases between 1944 and 1976. As a result, there is relatively little net increase/decrease in total radiative forcing and therefore, global surface temperature. The timing of these temperature effects is consistent with results obtained from model simulations (Andronova and Schlesinger, 2000; Tett et al., 1999)." The authors should reconcile their statement about the hiatus with the explanation based on a slight decrease in total forcing.

- **Added** a paragraph: "Kaufmann *et al.* (2006) note about the period 1944-76 that the decrease in net radiative forcing resulting from an increase in anthropogenic sulfur emissions approximately offset the increase in net forcing from an increase in WMGHGs. Aside from the uncertainty in measuring forcing from sulfur emissions, this offset nicely shows the relationship between the OMO and temperatures: while the OMO declined by an average of 0.016°C per year, temperature also declined by an average of 0.012°C per year. The decline in temperature over this period may be explained both by a decline in the OMO and an increase in sulfur emissions."

- We appreciate that the referee directed our attention to this. We believe that Kaufmann *et al.* (2006) may have inadvertently overstated their case. Interannual variations in temperature are large – precisely the reason we filter out high-frequency noise when calculating the OMO. A small but persistent movement in temperature that correlates with the OMO is obscured. Our point does not undermine the point raised by those author about the offset caused by the different types of anthropogenic emissions.

2. Page 5 Lines 1-3 describe the method used to covert a change in W/m2 to temperature $(0.536 + 0.561 \times 0.431 \approx 0.777, C)$. The authors should explain where 0.430 'comes from.'

- **Added:** "where 0.426 is the CCR estimate of $\alpha_1$ in the model in (2) with $\gamma, \delta = 0$ (see Table S.2 of the SOM)" [The 0.426 was 0.430, but these numbers have changed slightly.]

3. Page 11 "Our main findings for this period suggest that the three main factors driving the hiatus were (a) the unusually warm year of 1998, even conditional on the ENSO, (b) the ENSO itself, and (c) the increase in tropospheric aerosols during that period, though the latter is measured with a high degree of uncertainty" This is a very important component of the authors results, but these results are not really clear in the abstract. The authors should edit this abstract to make these results clearer. Also the abstract should highlight the result that the occurrence of a future hiatus depends on in part on the rate at which forcing grows.

- **Revised.** Thank you for this suggestion! We tried to execute this revision without lengthening the abstract too much.

**Literature cited**

[revised manuscript text omitted]

**S.1    Updating the Forcing Series**

We update the forcing series of Hansen *et al.* (2017) to 2016 as follows. We regress the first three series, $CO_2$, $CH_4$, $N_2O$, (in W/m$^2$) onto the natural log of the series given by NOAA[1] in ppm for carbon dioxide and in ppb for methane and nitrous oxide. We then predict the 2016  forcing using the natural log of the 2016 NOAA data. The 2016 solar forcing is updated by imposing the 2015-16 percent change from NASA.[2]

In contrast  to $CO_2$, $CH_4$, and $N_2O$, forcings from CFCs are changing very slowly and so we set 2016 to be the average of 2004-2015. We take the same moving average approach to estimate the 2016 forcings for ozone  and volcanoes as for CFCs. Volcanic activity over this period was not trivial, as noted by Vernier *et al.* (2011) and Neely *et al.* (2013), but neither are there any major eruptions on the order of Mount Pinatubo in 1991.

Forcings from tropospheric aerosols & surface albedo near the end of the sample are changing smoothly, but not as slowly as those from CFCs, ozone, and volcanoes, so we set the 2016 [3] forcing to be that of 2015 plus the average annual growth rate over 2004-2015.

**S.2    Estimating the Oceanic Multidecadal Oscillation**

Ocean cycles in mean temperature data – and the AMO in particular – have been estimated a number of ways in the literature. A key problem in estimating the cycle is removing the long-run trend due to global climate change. A common method for this purpose is linear detrending of GMT (Enfield *et al.*, 2001; Wyatt and Curry, 2014), although linear detrending has been criticized for this purpose by the IPCC (Bindoff *et al.*, 2013). Approaches using stochastic trends include those of Trenberth and Shea (2006), who use temperatures in other oceans to detrend the Atlantic, and Lenton *et al.* (2017), who use global mean temperature to detrend regions in the Atlantic and Pacific.
* * *
[1]Downloaded from www.columbia.edu/~mhs119/GHGs on May 15, 2017.

[2]Downloaded from solarscience.msfc.nasa.gov on May 15, 2017.

[3]

Another problem that we must avoid is over-fitting the statistical model in equation (2) in the paper. As an example of over-fitting, consider 2016, which was an unusually warm El Niño year. A variable constructed by simply detrending sea surface temperatures would have a particularly high value for 2016. Regressing GMT onto detrended GMT or sea surface temperatures would show a superficially good fit, in the sense that the model could not distinguish between secular cyclical variability and idiosyncratic noise.

Our approach handles these two problems by filtering out both long-run and short-run information from the time series of temperature anomaly distributions. We decompose the temperature anomaly $T^a$ into a long-run trend component $T^\tau$, a stationary multidecadally oscillating component $T^s$, and a noise component $T^n$, so that $T^a = T^\tau + T^s + T^n$. Similar to the AMO, we refer to $T^s$ as the Oceanic Multidecadal Oscillation (OMO).

To estimate the OMO, we first divide up the HadSST3 data into five ocean regions: North Atlantic ($i = 1$), South Atlantic ($i = 2$), North Pacific ($i = 3$), South Pacific ($i = 4$), and Indian ($i = 5$),  defined according to NOAA.[3] We then calculate the mean  temperature anomaly $\bar{T}^a_{it}$ for each region $i = 1, ..., 5$ at each year  $t = 1850, ..., 2016$ and then estimate a stochastic trend for each region. The trends are estimated by linear regressions of $\bar{T}^a_{it}$ onto a constant and WMGHGs  in W/m$^2$ relative to 1850 ($h^{WMGHG}_t$), which may be written as

$$\bar{T}^a_{it} = \alpha_{0i} + \alpha_{1i} h^{WMGHG}_t + v_{it}$$

for $i = 1, ..., 5$ and $t = 1850, ..., 2016$. They are identified by the predicted mean temperature anomaly $T^\tau_{it} = \hat{\alpha}_{0i} + \hat{\alpha}_{1i} h^{WMGHG}_t$ from each of these five regressions.

Our stochastic trend approach using WMGHGs is justified by a widely accepted physical link between temperatures and forcings from energy balance models like that in  equation (2) . In contrast, the linear deterministic trend approach replaces $h^{WMGHG}_t$ with $t$ in the regression above. In order for the linear trend approach to be justified by a physical model, forcings would have to increase by a constant amount over time, which is justified neither by the data nor by the process by which WMGHG emissions cumulate to concentrations. Moreover, varying coefficients over $i$ allow for heterogeneous multibasin climate sensitivity – i.e., it allows for the sea surface temperatures in different  ocean regions to be influenced by WMGHG differently over the long term.

Figure S.1 shows the predicted temperature trends $T^\tau_{it}$ from these regressions. Note that the Pacific and Indian Oceans share a common warming trend, while those of the Atlantic are quite a bit different. The Atlantic is warming faster than the other oceans, with the South Atlantic catching up to the Indian and Pacific from a colder starting point and the North Atlantic becoming increasingly warmer.
* * *
[3] See www.nodc.noaa.gov/woce/woce_v3/wocedata_1/woce-uot/summary/bound.htm.

[Figure]

**Figure S.1. Heterogeneous Oceanic Temperature Trends.** Fitted anomalies from regressing average temperature anomalies of five ocean regions onto a constant and WMGHGs.

In order to detrend the distribution of sea surface temperatures, we  subtract the estimated trend  $T_{it}^{\tau}$ for each region from the series of  temperature anomalies in each of the $5°$ by $5°$ boxes in that ocean.  Mathematically, this is $T_{k(i)t}^{s+n} = T_{k(i)t}^{a} - T_{it}^{\tau}$, where $k(i)$ indexes boxes in each ocean region $i$. Figure S.2 shows the actual number available for

5    each year of our sample. A maximum of $17,391$ is attained in 1979. Prior to the 1970's the number of observations generally increases over time, but with noticeable dips during major international disruptions, such as World War II, World War I, and the American Civil War.

    A standard nonparametric density estimation technique (Gaussian kernel with Silverman bandwidth) is used to estimate the density $f_t^{s+n}(r)$ of heterogeneously detrended sea surface temperature anomalies  $T_{kt}^{s+n}$, where $k$ now

10    indexes boxes across all ocean regions: in set notation, $\{T_{kt}^{s+n}\} = \cup_{i=1}^{5} \{T_{k(i)t}^{s+n}\}$. The detrending procedure above has removed $T_{it}^{\tau}$, so that $f_t^{s+n}(r)$ reflects the density of $T^s + T^n$ across all of the ocean boxes. We omit $0.5\%$ of the outliers in each tail, which we believe is an adequate threshold to ameliorate well-known boundary problems from kernel density estimation without substantively altering the moments of the distribution. The support of the density of outliers becomes $r \in [-2.995, 3.085]$. We

[Figure]

**Figure S.2. Number of Observations per Year from HadSST Data.** Number of observation available to estimate annual probability density functions.

abbreviate the endpoints of the support simply by $-$ for $r-$ and $+$ for $r^+$ throughout the SOM. Figure S.3 (top panel) shows the density $f_t^{s+n}(r)$ $f_t^{s+n}(r)$ of the stationary temperature distribution for each year.

Next, we smooth $f_t^{s+n}(r)$ to obtain $f_t^s(r)$ by removing short-run noise $T^n$. To do so this end, we first calculate the spatial mean $\int_-^+ r f_t^{s+n}(r) dr$, which could be referred to as the heterogeneously detrended oceanic mean temperature: it approximates the mean of $T_{kt}^{s+n}$ over $k$. We fit the result to a single sine function, estimating

$$\int_-^+ r f_t^{s+n}(r) dr = \theta_1 \sin(\theta_2(t/T) + \theta_3) + \theta_4 + e_t, \tag{S.1}$$

using nonlinear least squares.

Nonlinear least squares estimates a periodic function with an amplitude of $0.130°$C, a vertical shift of $-0.000°$C (nearly zero), a period given by $2\pi/13.75 \times T \simeq 76$ years, and a phase shift given by $0.21/13.75 \times T \simeq 3$ years (Table S.1). This period is roughly consistent with the Wyatt-Curry "stadium wave" with a half-period cooling regime of 31-38 years from about

[Figure]

**Figure S.3.**  **Unsmoothed and Smoothed Detrended Anomaly Distributions Over Time.** Top: Unsmoothed anomaly distribution from heterogeneously detrended sea surface temperature anomalies. Bottom: Smoothed anomaly distribution from fitting the unsmoothed distribution to a sine function, used to represent the OMO.

|            | Linear/Homog | | Trenberth-Shea | | WMGHG/Heterog | |
|------------|------|------|------|------|------|------|
|            | est. | s.e. | est. | s.e. | est. | s.e. |
| $\theta_1$ | 0.13 | 0.11 | 0.12 | 0.13 | 0.13 | 0.11 |
| $\theta_2$ | 14.66 | 2.71 | 13.53 | 4.99 | 13.75 | 2.95 |
| $\theta_3$ | $-0.11$ | 1.61 | 0.28 | 3.47 | $-0.21$ | 1.67 |
| $\theta_4$ | $-0.03$ | 0.08 | $-0.03$ | 0.10 | $-0.00$ | 0.08 |
| $R^2$ | 0.42 | | 0.34 | | 0.53 | |

**Table S.1. Periodic Function Estimation Results.** Results from fitting the  linearly/homogeneously detrended OMO,  the Trenberth-Shea (2006) AMO, and the WMGHG/heterogeneously detrended OMO to the  periodic function in (S.1).

[Figure]

**Figure S.4. Mean and Fitted Oscillations.** Mean and fitted oscillations using data detrended by a single linear trend, using  data from Trenberth and Shea (2006), and using data detrended by allowing heterogeneous WMGHG trends.

1940 to about 1975. The years over the sample in which the OMO has a neutral effect – neither cooling nor warming – are (approximately) 1852, 1928, and 2004.

As a comparison, Table S.1 and Figure S.4 compare the OMO and periodic function estimated in this manner with an OMO and periodic function estimated using linear detrending and with the AMO signal of Trenberth and Shea (2006)[4] and similarly estimated periodic function. The linear detrending method  yields a shorter period of 72 years, while the Trenberth-Shea AMO signal has a longer period of 78 years. Although the AMO signal has a longer period, the phase shift is negative, so

5    that the next peak occurs just after the end of the sample. In contrast, the linear detrending method shows a peak in about 2011 – in stark contrast to the recent high temperatures in 2015 and 2016. The table also shows $R^2$'s from the three regressions. These statistics should be interpreted with caution, both because they are $R^2$'s from nonlinear regressions and because the regressions have the same functional form of time but different regressands. Nevertheless, we interpret them as evidence that the periodic function is a better approximation to the oscillation estimated using the proposed detrending method.

10    Now, in order to  identify the distribution of $T^s$ from that of $T^s + T^n$, we construct a distribution that is changing only in its mean over time. To  this end, we first create a measure of the average distribution $f^n(r)$ of "de-cycled" anomalies $T^n$ with trend and periodic function removed. The density $f_t^{s+n}(r)$ is already detrended, but in order to remove the multidecadal cycle, we change the support by subtracting the heterogeneously detrended oceanic mean temperature $T^s = \hat{\theta}_1 \sin(\hat{\theta}_2(t/T) + \hat{\theta}_3) + \hat{\theta}_4$ estimated from equation (S.1) from each detrended temperature anomaly in each year. For example, the density

15    function estimated for a detrended temperature anomaly of $1°C$ in year $t$ becomes the "de-cycled" density function estimate at $(1 - T^s)°C$ and, most importantly, the density at $(T^s)°C$ becomes $0°C$. We then average the densities at each temperature anomaly $r$ across the sample, 1850-2016, obtaining a "de-cycled" density $f^n(r)$.

   Finally, we create a series of estimated densities  $f_t^s(r)$, smoothed versions of $f_t^{s+n}(r)$.  To that end, we reverse the procedure described above by adding the heterogeneously detrended oceanic mean temperature $T^s$ to $f^n(r)$

20    in order change the support back to  $r \in [-2.995, 3.085]$. We then remove outlying anomalies outside the original support. In this way, we estimate the smoothed density  $f_t(r) = f_t^s(r)$ of the OMO, displayed in the bottom panel of Figure S.3. The methodology clearly extracts a density that (a) appears to be stationary, devoid of a long-run stochastic or deterministic trend from warming, (b) appears to be smooth, devoid of idiosyncratic noise, and (c) appears to capture the multidecadal cycle.

   Disaggregated historical temperature data are measured with bias and uncertainty. By design, our method for smoothing the

25    cyclical component eliminates any idiosyncratic/short-run uncertainty in these data. If there is a static bias throughout the time span, it is picked up by a non-zero estimate of $\theta_4$ in equation (S.1), which we estimate to be nearly zero (Table S.1). Since we use $\theta_4$ to build the cyclical component, any bias in our estimate of $\theta_4$ is picked by the intercept $\alpha_0$ in equations (1) and (2), but should not affect the other coefficients estimates used to make our inferences. However, if the bias changes over time in a non-idiosyncratic way, modeling it would require a more involved strategy, which we leave for future research.

30    ## S.3    Energy Balance Model

We define location $\ell \in G = L \cup O$ where the set $G$ is all locations on the globe and $L$ and $O$ are sets of land and ocean locations. Location $\ell$ may be given as a latitude-longitude pair, in which case the integrals over $\ell$ below become double integrals over
* * *
[4]Downloaded from www.cgd.ucar.edu/cas/catalog/climind/AMO.html on July 17, 2017.

latitude and longitude. Letting the index $j = G, L, O$, $n_j = \int_j d\mu_\ell$ with counting measure $\mu$ denotes the number of locations in each set.

Adapting the energy balance model (EBM) of North (1975) and North and Cahalan (1981) to accommodate external forcing, we may write

$$C(\ell)dT_t(\ell) = QS(\ell)a(\ell) - (A + BT_t(\ell)) + D(\ell, T_t(\ell)) + h_t(\ell) + \varepsilon_t(\ell), \tag{S.2}$$

where $C$ is heat capacity, $T_t$ is temperature at time $t$, $Q$ is the solar constant, $S$ is solar irradiance, $a$ is co-albedo, $A + BT_t$ is emitted energy, $D$ is a linear-in-temperature approximation to the heat diffusion term in their model, $h_t$ is radiative forcing, and $\varepsilon_t$ is stochastic forcing. The stochastic forcing term is assumed by North *et al.* (1981) to be idiosyncratic, but we do not require this assumption.[5,6]

Historical temperature data sets typically express temperature in terms of anomalies from a base period in order to ameliorate well-known measurement errors. Accordingly, we decompose temperature as temperature during a base period $b$ plus the temperature anomaly, $T_t = T_b + T_t^a$. The two forcing components may also be decomposed into base plus anomaly, expressed as $h_t = h_b + h_t^a$ and $\varepsilon_t = \varepsilon_b + \varepsilon_t^a$. Adapting the EBM in (S.2) to accommodate temperature anomalies allows

$$(C + B)T_t^a - CT_{t-1}^a = D(\ell, T_t^a) + h_t^a + \varepsilon_t^a, \tag{S.3}$$

by subtracting $C(\ell)dT_b(\ell) = 0$ from both sides, discretizing the derivative to a unit increment, and suppressing the location argument for now.

Because $C, B > 0$, $\pi = C/(C + B) < 1$, so that the autoregressive component may be inverted. Doing so yields

$$T_t^a = (C + B)^{-1} \sum_{i=0}^{\infty} \pi^i [D(\ell, T_{t-i}^a) + h_{t-i}^a + \varepsilon_{t-i}^a] \simeq \frac{1}{B}[D(\ell, T_t^a) + h_t^a + \varepsilon_t^a],$$

where the approximation results from a Beveridge-Nelson-type decomposition (see Phillips and Solo, 1992). The approximation is more valid when the data are cointegrated or cotrending, in which case the neglected terms have a lower asymptotic order.

What does the stochastic forcing term $\varepsilon_t^a$ represent? Aside from noisy measurement of the data, it also accounts for otherwise missing components of temperature changes. Most notably missing are natural variability, such as changes in the ocean heat uptake, and other natural cycles, such as ENSO. As proxies, we employ the OMO, given by $T_t^s(\ell)$, and the SOI often used as a proxy for ENSO, given by $S_t$. Timmermann *et al.* (1999) note the possibility that external forcings may correlate with more frequent and/or severe ENSO cycles. By including the SOI in the model, we are implicitly assuming that the correlation is reflected in the SOI.

An alternative way to capture natural variability might be to allow for separate meridional ocean transport, along the lines of Rose and Marshall (2009), which could likely be accomplished along the lines of Pretis (2015) using a model that cointegrates
* * *
[5]North and Cahalan (1981) assume co-albedo to be a function of temperature as well as latitude, but more recent studies show that co-albedo is effectively constant in temperature at a given latitude (Stephens *et al.*, 2015; Stevens and Schwartz, 2012).

[6]Alexeev *et al.* (2005) note the effectiveness of modeling the diffusion coefficient $D$ as a function of temperature in order to capture polar amplification, which is not our aim.

surface temperature with deep ocean heat content. However, keeping in mind that our aim is to model hiatus periods that may be sparsely distributed over the historical record, the short time span over which ocean heat content is measured precludes this approach.

In such a model, surface temperatures, deep ocean heat content, and forcings share a single stochastic trend, so that the marginal value of the deep ocean heat content relative to forcings is natural variability. Hence, omitting deep ocean heat content does not cause a spurious regression, but rather relegates this stationary variability to the error term. Our proxies, the OMO and SOI, allow us to explicitly model the primary multidecadal and interannual sources of this variability.

Because the EBM does not explicitly include these indicators, it is natural to model them nonparametrically. To this end, we specify the model as

$$T_a = \alpha_0 + \alpha_1[h_t(\ell) + D(\ell, T^a(\ell))] + b(T^s_t(\ell)) + c(S_t) + \eta_t(\ell), \tag{S.4}$$

where

$$b(T^s_t(\ell)) = \begin{cases} \sum_{i=1}^{m_T} \gamma_i^T b_i(T^s_t(\ell)) \text{ for } \ell \in O \\ 0 \text{ for } \ell \in L \end{cases}$$

$$c(S_t) = \sum_{i=1}^{m_S} \gamma_i^S c_i(S_t)$$

are two generic series expansions intended to capture possibly nonlinear effects of these indicators. The OMO captures variability over the ocean, so this component is set to zero over land. The SOI is a single indicator. The last term $\eta_t(\ell)$ contains the original stochastic forcings and their lags and an allowance for finite-order approximation error of the two expansions. We may think of this term representing residual forcings, and it almost certainly exhibits temporal correlation.

Now we aggregate across locations to obtain a global model. To this end, let $O(T^s) = \{\ell \in O : T^s(\ell) = T^s\}$ be a subset of $O$ over which the $T^s(\ell)$ has the same numerical value, let $n_{O(T^s)} = \int_{O(T^s)} d\mu_\ell$ denote the number of locations in $O(T^s)$ that have the value $T^s$, and let $f$ be the probability density function of $T^s(\ell)$ in $O$ with support $[r^-, r^+]$. Note that

$$\int_-^+ b(r)f(r)ds = \int_-^+ \left[ n_{O(T^s)}^{-1} \int_{O(T^s)} b(T^s(\ell))d\mu_\ell \right] f(T^s)dT^s = n_O^{-1} \int_O b(T^s(\ell))d\mu_\ell,$$

which means we can aggregate all of the functions $b$ across ocean locations or we can aggregate all of the functions $b$ with the same observed argument and then aggregate them again with weights given by the frequency of each argument. Defining $T^a = n_G^{-1} \int_G T^a(\ell)d\mu_\ell$, $h = n_G^{-1} \int_G h(\ell)d\mu_\ell$, $\eta = n_G^{-1} \int_G \eta(\ell)d\mu_\ell$, integrating across locations, and noting that the diffusion term $D(\ell, T^a(\ell))$ is constrained to integrate to zero by the first law yields the EBM in equation (1) in the paper.

**S.4 Estimating the Energy Balance Model**

We approximate the functions $b$ and $c$ nonparametrically using a series of polynomial and trigonometric functions known as the flexible Fourier functional form, which Park *et al.* (2010) analyze using a semiparametric cointegrating regression much

| | OLS | CCR | | OLS | CCR | |
| | est. | est. | s.e. | est. | est. | s.e. |
|---|---|---|---|---|---|---|
| $\alpha_0$ | −0.315 | −0.313 | 0.018 | −14.254 | −13.881 | 13.130 |
| $\alpha_1$ | 0.413 | 0.426 | 0.020 | 0.431 | 0.440 | 0.015 |
| $\alpha_2$ | 0.079 | 0.081 | 0.026 | 0.054 | 0.055 | 0.021 |
| $\gamma_1$ | | | | 9.278 | 9.025 | 9.557 |
| $\gamma_2$ | | | | −8.687 | −8.446 | 9.705 |
| $\delta_1$ | | | | | | |
| $\alpha_0$ | −0.212 | −0.213 | 0.037 | −20.037 | −19.339 | 12.474 |
| $\alpha_1$ | 0.408 | 0.419 | 0.019 | 0.426 | 0.431 | 0.015 |
| $\alpha_2$ | 0.093 | 0.094 | 0.026 | 0.069 | 0.069 | 0.020 |
| $\gamma_1$ | | | | 13.540 | 13.043 | 8.370 |
| $\gamma_2$ | | | | −12.989 | −12.494 | 8.500 |
| $\delta_1$ | −0.055 | −0.054 | 0.017 | −0.062 | −0.061 | 0.012 |

**Table S.2. EBM Estimation Results.** Results from estimating the model in (2) in the paper using least squares (OLS) and asymptotically normal canonical cointegrating regression estimates (CCR) (Park *et al.*, 2010) with $p_1 = 0, 2$, $p_2 = 0, 1$, and $q_1, q_2 = 0$ in (S.4).

like ours. This form may be written as

$b_j^v(v) = v^j$ for $j = 1, ..., p_1$

$\qquad = \cos 2\pi k v$ for $j = p_1 + 2k - 1$ and $k = 1, ..., q_1$

$\qquad = \sin 2\pi k v$ for $j = p_1 + 2k$ and $k = 1, ..., q_1$

5 and analogously for $c_j^v(v)$, for $v \in [0, 1]$. Using this notation, $m_T = p_1 + 2q_1$ and $m_S = p_2 + 2q_2$. It is important that these functions are defined over the unit interval, so let $b_j(r) = (r^+ - r^-)b_j^v((r - r^-)/(r^+ - r^-))$ and $c_j(S) = (S^+ - S^-)c_j^v((S - S^-)/(S^+ - S^-))$, where $S^+$ and $S^-$ are the maximum and minimum observed SOI. Thus,

$$\int b_j(r) f_t(r) dr = \int (r^+ - r^-)b_j^v((r - r^-)/(r^+ - r^-)) f_t(r) dr$$

holds, making estimation convenient by simply multiplying $b_j^v$ by the range of values in the domain of the OMO.

10 The optimal orders $(p_1, q_1, p_2, q_2) = (2, 0, 1, 0)$ – i.e., $m_T = 2$ and $m_S = 1$ – are jointly determined by Schwarz-Bayesian and Hannan-Quinn information criteria evaluated using least squares with $p_1, p_2$ up to 3 and $q_1, q_2$ up to 2. With $m_S = 1$, SOI enters linearly and the regressor is thus simply $(S_t - S^-)$.[7] The estimation results are given in Table S.2.
* * *
[7]All models are expected to be cointegrated, because we simply add stationary series to the model with $(p_1, q_1, p_2, q_2) = 0$, which we found to be stationary cointegrating (see footnote 1 in the paper). More formally, we run residual-based augmented Dickey-Fuller tests which strongly rejected reject the null of no cointegration up to four lags.

| | $\underset{\sim}{v^2}$ $\underset{\sim}{\cos 2\pi k v}$ $\underset{\sim}{°C}$ | $\underset{\sim}{v^1}$ $\underset{\sim}{v^3}$ $\underset{\sim}{\sin 2\pi k v}$ $\underset{\sim}{\%\ \text{missing heat}}$ |
|---|---|---|
| Coefficient Estimates | | |
| Aggregate Effects | | |
| $\underset{\sim}{(p,q)=(1,0)}$ | | 0.75 |
| | 0.122 | −47.6 |
| $\underset{\sim}{(p,q)=(2,0)}$ | | 13.54 |
| $\underset{\sim}{−12.99}$ | | |
| | 0.113 | −44.1 |
| $\underset{\sim}{(p,q)=(3,0)}$ | | −176.56 |
| $\underset{\sim}{447.34}$ | | $\underset{\sim}{−335.76}$ |
| | 0.092 | −35.9 |
| $\underset{\sim}{(p,q)=(2,1)}$ | | 1236.48 |
| $\underset{\sim}{−1502.58}$ | | |
| $\underset{\sim}{103.87}$ | | $\underset{\sim}{−60.31}$ |
| | 0.029 | −11.3 |

**Table S.3. Robustness Checks on Order of the OMO Expansion.** Results from estimating the model in (2) in the paper using least squares with $p_1 = 1, 2, 3$, with $q = 0$ fixed and $q_1 = 0, 1$ with $p_1 = 2$ fixed in (S.4).

The estimate of the aggregate effect of the OMO (net of forcings) reported in the paper for the optimal order, $(p_1, q_1) = (2,0)$, is $0.11°C$ $(0.08, 0.14)°C$ over 1998-2013, or a $41.7\%$ $(30.0, 53.5)\%$ exacerbation of the puzzle of the missing heat. As a robustness check on the effect of selection of $p_1$ and $q_1$, we varied each $\pm 1$ from the optimal order: $(1,0), (3,0), (2,1)$, yielding estimates of the aggregate effects of $0.122°C$ ($47.6\%$ exacerbation), $0.092°C$ ($35.9\%$ exacerbation), and $0.029°C$ ($11.3\%$ exacerbation), by way of least squares. The estimates with $(1,0)$ or $(2,0)$ seem the most plausible, because the additional terms of models with $(3,0)$ and $(2,1)$ generate estimates with very large magnitudes and opposing signs, a classic sign of near multicollinearity.

**S.5 Estimation of Uncertainties**

**S.5.1 Uncertainties from Estimating the OMO**

Estimating the OMO relies on a statistical approximation, and we employ a parametric bootstrap strategy similar to that of Poppick *et al.* (2017) to account for uncertainty in estimation. Specifically, after fitting the periodic function in (S.1), we fit the residuals to an AR(1) and redraw from the residuals of the fitted AR(1). We re-create the regressand using a re-created AR(1) error with fitted autoregressive parameter (0.49). We then re-estimate all the parameters of the nonlinear regression. We conduct 999 bootstrap replications in this manner, and the sample paths plotted in Figure 2  of the paper reflect the periodic functions with 0.05 and 0.95 quantiles of $\theta_2$ in (S.1), which determines the period.

Our bootstrap differs from a typical bootstrap in that we do not re-estimate only the period and restrict the other parameters to their fitted values. Re-estimating all of the parameters allows uncertainty about the period to be correlated with uncertainty about the phase shift in particular, because uncertainty in both parameters affects our dating of the function's optima.

**S.5.2 Approximations to Uncertainties in Forcings**

Myhre *et al.* (2013) estimate forcings (in W/m$^2$  with 90% confidence from WMGHGs as $2.83 \pm 0.29$, or $\pm 10\%$. Volcanic forcings are estimated to be $-0.06$ ($-0.08$ to $-0.04$) over 1999-2002 and $-0.11$ ($-0.15$ to $-0.08$) over 2008-2011, or $\pm 33\%$. Similarly, forcings from solar irradiance are estimated to be $-0.05 \pm 0.05$, or $\pm 100\%$. The data from Hansen *et al.* (2017) for ozone are calculated as the simple average of forcings from tropospheric and stratospheric ozone (see footnote c of their Table A1). The total is estimated by Myhre *et al.* (2013) ~~state that uncertainty about forcing from WMGHGs is about ±10%, and indeed this roughly corresponds to the numerical values given for 2010. The numerical values given for volcanic forcings over the two periods correspond to ±33% within rounding error. If we make a similar assumption regarding the other forcings , then uncertainties about forcings from ozone, tropospheric aerosols & surface albedo , and solar are ±70%, ±143%, and ±100% respectively in 2010.~~ to be $0.35 \pm 0.20$, so that the average is $0.175 \pm 0.10$, or $\pm 57\%$. Myhre *et al.* (2013) estimate forcings from changes in surface albedo as $-0.15 \pm 0.10$ and from tropospheric aerosols as $-0.9$ (–1.9 to –0.1). Recent data from Hansen *et al.* (2017) put the total close to $-1.15$, so we consider the uncertainty to be $-1.15 \pm 1.00$, where $1.00 = 0.10 + (-0.1 - (-1.9))/2$, or $\pm 87\%$.

We assume that these percentages are roughly the same each year, as they are for WMGHGs and appear to be for volcanoes. In this way, we generate intervals for each forcing in each year to be consistent with the given data. This procedure inherently but realistically allows heteroskedasticity, because the uncertainty grows as the forcing's value grows. However, it does not allow for heteroskedasticity due to more precise measurements over time,  as noted by Myhre *et al.* (2013) for solar, e.g. We further assume that the uncertainties in the forcings are purely idiosyncratic in the sense that they are neither time-dependent nor mutually dependent.

Approximating the distributions of these uncertainties by a Gaussian distribution centered at the reported value, we can estimate the variance by dividing the difference in the quantiles by $2 \times 1.645$ and squaring the result, which reverses the

formula $\pm 1.645\sqrt{\mathbf{var}(v_t)}$ to calculate 90% intervals from the variance of a mean-zero Gaussian random variable $v_t$. Doing so generates a $5 \times 5$ variance/covariance matrix for each time period, with the estimated variance of each of the five forcings along the main diagonal and zeros elsewhere.

The average of the diagonals, reflecting the variances of the uncertainty for each forcing (WMGHGs, ozone, aerosols, solar, and volcanic respectively), is estimated to be $(0.007,$  $0.001, 0.105, 0.002, 0.012)'$ over 1850-2016 and  $(0.029, 0.002, 0.359, 0.004, 0.000)'$ over 1999-2013, the recent fifteen-year hiatus period. As expected, forcings from aerosols are estimated to be the most uncertain. The uncertainties over the hiatus period are generally larger than those over the whole sample, reflecting the larger magnitudes of the forcings near the end of the sample. In contrast, volcanic forcings are nearly zero during the hiatus period, reflecting the absence of a volcanic eruption with a major impact on global climate.

**S.5.3 Contribution of Regressor Uncertainty to Coefficient Estimators**

Uncertainty in the regressors may be treated as "classical measurement error" in the parlance of the econometrics literature, which is known to cause bias in the coefficient estimates. Because we observe forcings with error, we may denote our observation of forcings by $h_t = h_t^0 + v_t$, where $h_t^0 = (h_{1t}^0, h_{2t}^0)'$, with $h_{1t}^0$ and $h_{2t}^0$ denoting respectively non-volcanic and volcanic forcings if the forcings could be observed without uncertainty. Similarly, $v_t = (v_{1t}, v_{2t})'$ such that $v_{1t}$ and $v_{2t}$ are respectively the sum of uncertainties about non-volcanic forcings, estimated as described above, and uncertainty about volcanic forcing. $v_t$ has a mean of zero and its components have variances given by $\sigma_{v1,t}^2$ and $\sigma_{v2,t}^2$. The former is the sum of the variances of the non-volcanic forcings, as the covariances are assumed to be zero.

In a cointegrating model like ours, bias in the long-run relationship is not hard to fix. In fact, although it was not designed to do so, the feasible Canonical Cointegrating Regression (CCR) methodology of Park *et al.* (2010) already takes into account this bias. Using a closely related model, Miller (2010, Theorem 2) shows the CCR estimator to be consistent, asymptotically normal, and asymptotically unbiased, with a variance that takes into account the measurement uncertainty.[8]

**S.5.4 Contribution of Uncertainty to Forecasts**

Explaining the contributions of the uncertainty to the missing heat of the 1998-2013 episode requires a measure of in-sample fit of $T_t^a$ for some arbitrary time period $t = 0$, given by  $\hat{T}_0^a = h_0^{*\prime}\hat{\alpha} + x_0'\hat{\gamma} + w_0\hat{\delta}$. For simplicity, denote the right-hand side by $z_0'\hat{\pi}$ with  $z_t = (h_t^{*\prime}, x_t', w_t)' = (1, h_{1t}, h_{2t}, x_t', w_t)'$ and $\pi = (\alpha', \gamma', \delta)' = (\alpha_0, \alpha_1, \alpha_2, \gamma', \delta)'$ and let $z_t^0 = (1, h_{1t}^0, h_{2t}^0, x_t', w_t)$ and $\varpi_t = (0, v_{1t}, v_{2t}, 0, 0)$, such that $z_t = z_t^0 + \varpi_t$. The variance of the uncertainty in  $\hat{T}_0^a$ is given by

$$\mathbf{var}(\hat{T}_0^a | z_0^0) = z_0^{0\prime}\mathbf{var}(\hat{\pi}|z_0^0)z_0^0 + \mathbf{var}(\varpi_0'\hat{\pi}|z_0^0) + 2z_0^{0\prime}\mathbf{cov}(\hat{\pi}, \varpi_0'\hat{\pi}|z_0^0)$$
* * *
[8]We do not model uncertainty in measuring temperatures.  Nor do  we model the effect of uncertainty in the volcanic forcings on the coefficient estimate, because the uncertainty is much smaller than the uncertainty for the other forcings – nearly zero – over the 1998-2013 period.

using this notation.

If we could observe $z_0^0$, a (Gaussian) 90% uncertainty interval for $\hat{T}_0^a$  $\hat{T}_0^a$ would be

$$z_0^{0\prime}\hat{\pi} \pm 1.645\sqrt{z_0^{0\prime}\mathbf{var}(\hat{\pi}|z_0^0)z_0^0}$$

from the first term. Instead, rewrite the variance as

$$\quad \mathbf{var}(\hat{T}_0^a|z_0^0) = z_0^{0\prime}\mathbf{var}(\hat{\pi}|z_0^0)z_0^0 + \mathbf{var}(\hat{\alpha}_1|z_0^0)\bar{\sigma}_{1v}^2 + \mathbf{var}(\hat{\alpha}_2|z_0^0)\bar{\sigma}_{2v}^2$$
$$+ \left[2z_0^{0\prime}\mathbf{cov}(\hat{\pi},\varpi_0'\hat{\pi}|z_0^0)\right] + \left[\mathbf{var}(\varpi_0'\hat{\pi}|z_0^0) - (\mathbf{var}(\hat{\alpha}_1|z_0^0)\bar{\sigma}_{1v}^2 + \mathbf{var}(\hat{\alpha}_2|z_0^0)\bar{\sigma}_{2v}^2)\right]$$

where $\bar{\sigma}_{1v}^2$ and $\bar{\sigma}_{2v}^2$ are temporal averages that estimate the variances $\sigma_{v1,t}^2$ and $\sigma_{v2,t}^2$ at $t = 0$. Specifically, we use the averages over 1999-2013 given above, so that $\bar{\sigma}_{2v}^2$ is effectively zero.

Ignoring the two terms in brackets and setting $\bar{\sigma}_{2v}^2 = 0$, a (Gaussian) 90% uncertainty interval for $\hat{T}_0^a$ given by

$$\quad z_0'\hat{\pi} \pm 1.645\sqrt{z_0^{0\prime}\mathbf{var}(\hat{\pi}|z_0^0)z_0^0 + \mathbf{var}(\hat{\alpha}_1|z_0^0)\bar{\sigma}_{1v}^2}$$

takes into account uncertainty in the non-volcanic forcings. Because the uncertainty in the regressors is correlated with the uncertainty in the estimator, the bracketed terms are not zero, but we expect that they will be small.

As with predictions from any linear model, the variance of the out-of-sample conditional forecasts is augmented by the estimated variance of the error term $\eta_t$. In that case, we drop $\mathbf{var}(\hat{\alpha}_1|z_0^0)\bar{\sigma}_{1v}^2$, because we are conditioning on specific data

15 and there are no measurement errors, and  we use least squares rather than CCR, in order to minimize mean squared forecast error.

**S.6   Results with an Alternative Measure of Aerosol Forcing**

As noted in the paper, the data on forcings from tropospheric aerosols are particularly uncertain and the data from Hansen *et al.* (2017) are intentionally smoothed. As an alternative data source, we consider the data on anthropogenic emissions of sulfur

20 dioxide from Hoesly *et al.* (2018). Because sulfur dioxide has such a short residence time in the atmosphere, the natural log of emissions should be a reasonably good proxy for forcings within a year, in contrast to well-mixed greenhouse gases, the residence times of which require the natural log of concentrations cumulated from emissions in order to approximate forcings.

To convert the log of emissions into a measure in W/m$^2$, we regress the series of forcings from tropospheric aerosols and surface albedo of Hansen *et al.* (2017) onto an intercept and the log emissions series. The fitted prediction from this regression

25 is the linear transformation of log emissions data that best approximates the scale of the forcings from both tropospheric aerosols and surface albedo, and it may be expressed in W/m$^2$ as desired. Figure (S.5) shows a comparison of the two series, as well as the uncertainty interval calculated for the former series using $\pm 87\%$ as discussed above. The magnitude of the uncertainty clearly allows for either measure.

Table (S.4) shows our key results for the hiatus calculated with the two measures. All of the results in the paper are calculated

30 using the data from Hansen *et al.* (2017), but we also report in the paper the contribution of tropospheric aerosols and surface

| Contributions (°C) from: Hansen et al. (2017) Hoesly et al. (2018) | |
|---|---|
| Volcanoes | −0.003 |
| (−0.001, −0.004) | −0.002 |
| (−0.001, −0.004) | |
| TASA | −0.050 |
| (−0.030, −0.070) | 0.003 |
| (−0.014, 0.020) | |
| Solar | 0.016 |
| (0.013, 0.018) | 0.014 |
| (0.012, 0.017) | |
| OMO | 0.110 |
| (0.079, 0.141) | 0.083 |
| (0.050, 0.117) | |
| SOI | −0.062 |
| (−0.029, −0.095) | −0.063 |
| (−0.030, −0.097) | |
| Model in (2) with $\gamma, \delta = 0$ | −0.178 |
| (−0.155, −0.202) | −0.108 |
| (−0.085, −0.131) | |
| Model in (2) with $\gamma, \delta \neq 0$ | −0.110 |
| (−0.072, −0.147) | −0.068 |
| (−0.023, −0.113) | |

**Table S.4. Estimated Contributions of Key Potential Explanations of the 1998-2013 Episode.** Negative temperatures help to explain the missing heat, while positive temperatures exacerbate the puzzle. Tropospheric aerosols & surface albedo are abbreviated by TASA. 90% confidence intevals are shown in parentheses.

[Figure]

**Figure S.5. Alternative Estimates of Forcings from Tropospheric Aeresols and Surface Albedo.** Original forcings from Hansen *et al.* (2017) plotted with 90% uncertainty intervals of $\pm 87\%$ against estimated forcings from Hoesly *et al.* (2018).

albedo (TASA) using the data calculated from that of Hoesly *et al.* (2018). Because the alternative measure of TASA is aggregated with the other non-volcanic forcings in the model in (2), all of the estimates are influenced by the choice of TASA forcing measure. That of the contribution of TASA obviously is influenced the most. The difference in effect is so large that the intervals do not overlap and the direction of the effect of TASA in explaining the hiatus is effectively unknown – it may

5  exacerbate rather than help to explain the hiatus.

A statistical mystery remains. The two measures are apparently within measurement error of each other in Figure (S.5), and stationary measurement error does not affect asymptotic inference about cointegration relationships. We have constructed our intervals to allow for stationary measurement error over 1998-2013, and we report cointegration diagnostics in footnote 1 of the paper. The diagnostics are not substantively different using the alternative measure, suggesting that the measurement error is in

10  fact stationary and that both estimates are consistent. So, we should expect very similar estimates of $\alpha_1$ in (2) and thus similar estimates of the contribution of TASA in Table (S.4). Yet they are not similar. Moreover, Figure (S.5) suggests that, in spite of the diagnostics, the difference between measures may not be stationary. We leave further investigation of this discrepancy for future research.

---

## Author Response (AR3)

**University of Missouri**
Department of Economics
College of Arts and Science

**J. Isaac Miller**

millerjisaac@missouri.edu

221 Professional Building
Columbia, MO 65211-6040

September 22, 2020

*Earth System Dynamics*
Fubao Sun, Handling Editor

Dear Dr. Sun:

Please consider the **revised** manuscript, entitled "Dating Hiatuses: A Statistical Model of the Recent Slowdown in Global Warming -- and the Next One" (with Kyungsik Nam), for publication in *Earth System Dynamics*.

We have carefully and thoughtfully addressed the reviewers' comments, which we found to be constructive and certainly helped us to improve the draft, in the response and in the manuscript. A response to each reviewer follows. Because we have received positive feedback from four reviewers and this is our fourth submission (third revision), we sincerely hope that you can make a quick acceptance decision!

Sincerely,

J. Isaac Miller

**Authors' Response to the reports of Reviewer #3 (Report #1) and Reviewer #4 (Report #2) on the 3rd submission of "Dating Hiatuses: A Statistical Model of the Recent Slowdown in Global Warming -- and the Next One"**

**Reviewer #3 (Report #1):**

I thank the authors for their careful attention to my comments. They are addressed satisfactorily in the revised manuscript. Indeed, their manuscript successfully marries sophisticated statistical analysis with a sophisticated understanding of the science. As such, I believe the manuscript should be published.

- We appreciate the referee's time in helping us improve the manuscript!

I do have one last suggestion for the final manuscript. On pages 6-8, the authors repeatedly refer to the absolute and percentage of the missing heat that is associated with a given variable. For example, on page 6 (lines 9 – 10) they argue that volcanoes account for about 1.1 % of the missing heat and again on page 7 (lines 34-35) "that ENSO explains 23.7% of the missing heat. I suggest that the authors create a figure that allows readers to see these absolute amounts or percentages in one place. This will allow reader to compare various contributions. I realize that many of these findings are implicit in Figure 1. But the slopes of the temperature lines are much harder to quantify and interpret relative to a bar chart that explicitly shows the quantity of missing heat that is associated with volcanoes, ENSO, etc.

- **Added a bar chart** with the suggested information and some additional explanatory remarks. Thank you for the suggestion!

Also figure 1 does not show the change in temperature previous studies attribute to various factors. Indeed, the bar chart could also show the quantity of 'missing heat' that previous analyses attribute to a particular variable, such as volcanoes or ENSO. When possible, this would help readers compare the authors' results to those generated by previous studies, which the authors review on pages 1-2.

- The principle underlying this comment is sound, but unfortunately the implementation of this well-intentioned comment proved to be too difficult.
- It would be impossible, or nearly so, to compare the denominator, missing heat, across papers that define missing heat differently. If nothing else, many of the cited papers were published prior to our dating of the end of the hiatus in 2013, so they could not define it in the same way. Even those papers published later may not have defined the hiatus in the same way or over the same period (Schmidt et al., 2014; Karl et al., 2015; Yao et al., 2016; and Medhaug et al., 2017), as we wrote in the third paragraph.
- We considered trying to compare percentages instead, even though the denominators (missing heat) may be different. But not all of the preceding papers include calculations of missing heat and may present the hiatus differently. We thought it wiser to let the reader draw her/his own comparisons rather than trying to back out an object that the authors of those papers may not have had in mind.

**Reviewer #4 (Report #2):**
My concern is that the variable modelled and predicted in this study is subject to short- or long-term memory.

- **Added a clarifying sentence** to the sixth paragraph of the introduction: "In doing so, we carefully decompose the distribution of temperature anomalies into components with long memory, low-frequency, stochastic trending behavior (mapped to forcings from WMGHGs), with short memory, medium-frequency, cyclical behavior (the OMO), and with short memory, high-frequency, idiosyncratic behavior."
- We appreciate this very short yet very appropriate comment. We treated this issue carefully and extensively in our methodology, as described in the SOM. However, we acknowledge that this would not be clear to a time-series-inclined reader who chooses not to read the SOM. Although we did make it clear that we were using a methodology specifically aimed at series with stochastic trends, we did not make it clear in the main text that our measure of the OMO filters out both low and high frequencies. This should now be clear.

[We did not see any further comments in Report #2.]